# Characterization of viscoelastic behavior of basalt fiber asphalt mixtures based on discrete and continuous spectrum models

**Qi Huang[1], Xiaogang Kang[2], Pengfei Chen[3], Zhengji Zhang[2], Er-hu Yan[4]\*, Zhuohang Zang[2], Han Yan[5]**

1 School of Civil Engineering and Communication, North China University of Water Resources and Electric Power, Zhengzhou, P. R. China, 2 School of Civil Engineering and Communication, North China University of Water Resources and Electric Power, Zhengzhou, China, 3 CCCC First Highway Engineering Group Co., Ltd., Beijing, China, 4 Research Institute of Highway, Ministry of Transport, Beijing, China, 5 School of Traffic and Transportation Engineering, Changsha University of Science and Technology, Changsha, China

\* 13781229603@163.com

**Data Availability Statement:** All relevant data are within the paper.

**Funding:** The author(s) received no specific funding for this work.

## Abstract

In order to analyze the differences between the master curves of relaxation modulus $E(t)$ and creep compliance $J(t)$ obtained from discrete and continuous spectrum models, and to comprehensively evaluate the effect of basalt fiber content on the viscoelastic behavior of asphalt mixtures, complex modulus tests were conducted for asphalt mixtures with fiber content of 0%, 0.1%, 0.2% and 0.3%, respectively. Consequently, the master curves of Viscoelastic Parameters of asphalt mixtures were constructed according to the generalized Sigmoidal model(GSM) and the approximate Kramers-Kronig (K-K) relationship. Then, transformation of master curves using discrete and continuous spectrum models to obtain the models of $E(t)$ and $J(t)$ containing all viscoelastic information. Also, the accuracy of the models of $E(t)$ and $J(t)$ was evaluated. The results show that the addition of basalt fibers improves the strength, stress relaxation and deformation resistance of asphalt mixtures. It is worth noting that basalt fibers achieve the improvement of asphalt mixtures by changing their internal structure. Considering the different viscoelastic master curves at four dosages, the optimum fiber dosage was 0.2%. In addition, both discrete and continuous model conversion methods can obtain high accuracy conversion results.

## 1 Introduction

Cracking at low temperatures in winter and overloaded plastic deformation at high temperatures in summer are the most common distresses of asphalt pavements [1, 2]. With further loads and environmental effects, these distresses can develop into potholes and ruts, significantly reducing the service life and service quality of the pavement [3, 4]. Some studies have found that the incorporation of short-cut fibers into asphalt mixtures can improve the structure of asphalt mixtures, increase their tensile properties and deformation resistance, and thus enhance the low-temperature cracking resistance and high-temperature stability performance

**Competing interests:** The authors have declared that no competing interests exist

of pavements [5]. Short-cut basalt fibers and short-cut polymer fibers are the most commonly used road fibers. Compared with the latter, basalt fibers have better high temperature stability and mechanical properties [6, 7]. Many researchers have studied the viscoelastic behavior of basalt fiber asphalt mixtures, and they have used more dynamic modulus and phase angle to analyze its viscoelastic behavior, and seldom evaluate their viscoelastic properties comprehensively [8, 9].

Dynamic modulus $|E^*(\omega)|$, storage modulus $E'(\omega)$, loss modulus $E''(\omega)$, relaxation modulus $E(t)$, and creep compliance $J(t)$ are commonly used to describe the viscoelastic behavior of asphalt mixtures [10]. Different viscoelastic parameters characterize the different viscoelastic response. Among them, $E'(\omega)$ and $E''(\omega)$ are all functions with respect to frequency and describe the strength properties, purely elastic behavior and purely viscous behavior of the asphalt mixtures, respectively [11]. $E(t)$ and $J(t)$ are all functions of time and describe the stress relaxation and deformation capacity of the asphalt mixture [12]. However, the cost of obtaining accurate $E(t)$ and $J(t)$ is significant due to factors such as test instrumentation, test time and test operability. In contrast, complex modulus tests are simpler to operate and the tests are more easily controlled, so it is widely used to evaluate the viscoelastic properties of mixes [13].

Time spectra are powerful tools to realize the transformation of viscoelastic parameter master curves of asphalt and asphalt mixtures, they are the core of viscoelastic parameter master curve transformation. Through them it is possible to test the transformation of viscoelastic parameter master curves in different domains. Discrete model and continuous model time spectra are commonly used methods for viscoelastic master curve conversion. There are two main drawbacks of using the above discrete and continuous spectrum models for the master curve conversion of viscoelastic parameters [14–16]. First, the conversion process ignores the information provided by loss modulus and loss compliance, and the constructed the master curves of $E(t)$ and $J(t)$ do not contain all the viscoelastic information. Secondly, fitting the models of $E(t)$ and $J(t)$ separately increases the number of fits and reduces the accuracy of the converted master curves.

The approximate Kramer-Kronig (K-K) relationship can be used to build the dynamic modulus and phase angle master curves [17]; H. Liu utilized the approximate K-K relationship to construct the master curve model of storage modulus and loss modulus, and the study showed that the approximate K-K relationship can represent the interrelationships between the viscoelastic parameters very well [15];Yu D utilized the approximate Kramer-Kronig relationship between energy storage modulus and loss modulus to represent the loss modulus master curve, which is a simple and convenient process [18]. W. Luo found that the symmetric sigmodial model only describes the symmetric dynamic viscoelastic properties, it cannot accurately describe the asymmetric dynamic viscoelastic properties of asphalt mixtures [19]; Abtahi S M found that the master curve constructed by the asymmetric generalized Sigmoidal model is smoother by comparing several different Sigmoidal models [3]; The complex modulus master curve of asphalt mixtures established by Tan G. using an asymmetric generalized Sigmoidal model(GSM), which is able to quasi-responsive to the dynamic mechanical response with high correlation coefficients and accurately predicts dynamic mechanical properties [20]; Dynamic modulus principal curves and phase angle principal curves were constructed by the approximate K-K relationship and the GSM by F. Zhang [21]. The simultaneous application of both models ensures that the plotted principal curves conform to the line viscoelasticity, allowing for principal curves that are not symmetric about the point of view.

In view of the above problems, to evaluate the difference of the master curve obtained by the transformation of discrete and continuous models, and to achieve the purpose of comprehensively characterizing the viscoelastic behavior of basalt fiber asphalt mixtures. In this study, the approximate K-K relationship and the GSM are applied to construct the master curves of

dynamic modulus, phase angle, storage modulus and loss modulus. Then, the master curves of $E(t)$ containing all viscoelastic information are constructed from the master curves of $E'(\omega)$ and $E''(\omega)$ using discrete and continuous spectrum models, respectively. Next, the master curves of $J(t)$ is derived from the integral relationship between the $E(t)$ and $J(t)$ in the time domain. Finally, the accuracy of the master curves obtained from the discrete and continuous spectrum models is evaluated.

## 2 Materials and test methods

### 2.1 Materials

Basalt fiber asphalt mixtures are mostly used in the upper and middle surface layers of roads, therefore, the type of asphalt mixtures used in this study was AC-13 dense grading asphalt mixtures, as shown in Fig 1. The asphalt used No. 70 base asphalt produced in Tianjin. The coarse and fine aggregates and mineral powder come from limestone produced in Tongzhou. Basalt fiber chose Changsha North America floating company produced 9mm short cut basalt fiber, its length to diameter ratio of 560. In this study, the fiber content was selected as 0%, 0.1%, 0.2% and 0.3% (the fiber content accounted for the mass fraction of the asphalt mixture), and for the convenience of the study, they were numbered as BF-0, BF-1, BF-2 and BF-3, respectively. The basalt fiber asphalt mixture is prepared by dry mixing method to ensure that the basalt fiber is fully dispersed in the asphalt mixtures.

### 2.2 Complex modulus tests

According to the AASHTO TP 79–12, the complex modulus test was carried out using the asphalt mixture performance tester (AMPT) produced by the Italian Matester Company [22]. The test temperature is -10°C, 20°C and 35°C. The dynamic modulus and phase angle of six loading frequencies of 25Hz, 10Hz, 5Hz, 1Hz, 0.5Hz, and 0.1Hz were measured at each temperature, and the loading sequence was sequentially loaded from high frequency to low frequency.

## 3 Theoretical background

### 3.1 Complex modulus test parameters

The complex modulus $E^*$ dynamic modulus $|E^*|$, storage modulus $E'$, loss modulus $E''$, and phase angle $\varphi$ have the relationships shown in Eqs (1)–(4):

$$E^* = E' + iE''  \tag{1}$$

$$|E^*| = \sqrt{(E')^2 + (E'')^2}  \tag{2}$$

$$E' = |E^*|\cos\varphi  \tag{3}$$

$$E'' = |E^*|\sin\varphi  \tag{4}$$

### 3.2 The Time-Temperature Superposition Principle (TTSP)

The viscoelasticity of asphalt mixture has a strong dependence on time and temperature, and its viscoelastic behavior shows the equivalence of time and temperature, which is called TTSP

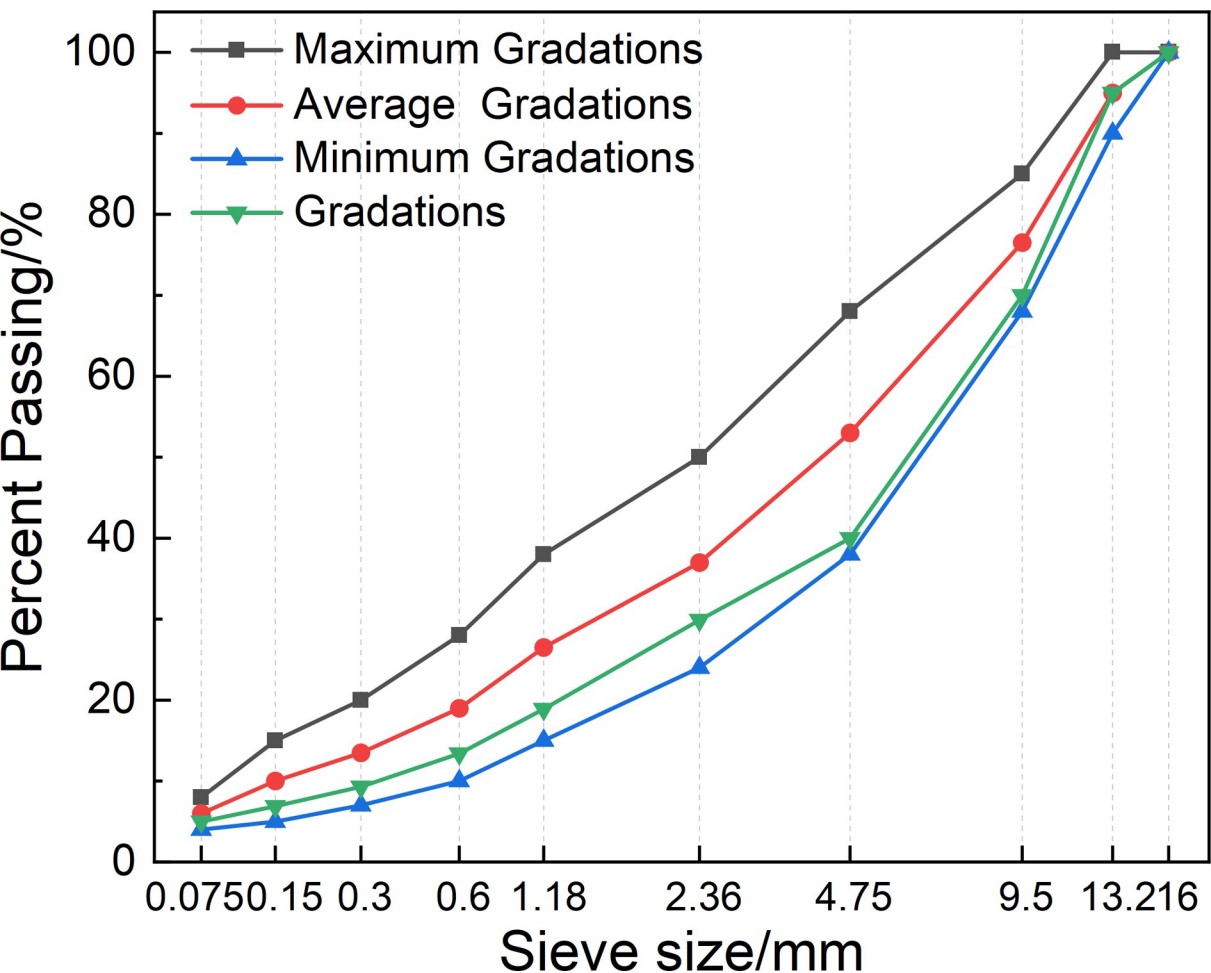

**Fig 1. Basalt fiber asphalt mixtures gradation.**

[23]. When TTSP is applied, the test curves at different temperatures can be moved to the target temperature curve, and the moving distance is called the shift factor $\alpha_T$. In this paper, when constructing the master curves the Williams-Landel-Ferr (WFL) equation shown in Eq (5) is chosen to calculate $\alpha_T$, because its parameters have clear physical meaning and the calculation results are more accurate [24].

$$\log \alpha_T = \frac{-C_1(T - T_r)}{C_2 + (T - T_r)}$$

(5)

Where $C_1, C_2$ is the material constant; $T$ is the test temperature; and $T_r$ is the reference temperature.

### 3.3 Construction of master curves in frequency domain

In order to ensure that the constructed the master curves of $|E^*(\omega)|$ and $\varphi(\omega)$ both conform to linear viscoelasticity and allow the master curves to be unsymmetrical about the viewpoint, the (GSM)and the approximate K-K relationship are used [25]. Like the dynamic modulus, the shape of the principal curves of the storage modulus can be described by a generalized

Sigmoidal model, and then using the approximate K-K relationship between the storage modulus and the loss modulus, the loss modulus principal curve model is derived [18].

## 3.4 Determining the discrete spectrum models

As the most commonly used discrete spectrum, the Prony series model is widely used to characterize the linear viscoelastic behavior of asphalt mixtures. The discrete spectrum composed of the coefficients of the Prony series contains all the viscoelastic information. The discrete spectrum can organically combine the viscoelastic parameters in different domains, and then realize the transformation of the master curve of the viscoelastic parameters.

**3.4.1 Determining the discrete relaxation spectrum.**   According to the Boltzmann superposition principle, the integral constitutive relation of the $E(t)$ and $J(t)$ is [26]

$$\int_0^t E(t - \tau) \frac{dJ(\tau)}{d\tau} d\tau \tag{6}$$

From the GMM and the GKM, the Prony series expression of $E(t)$ and $J(t)$ can be deduced as follows [27]:

$$E(t) = E_e + \sum_{i=1}^m E_i e^{-(t/\rho_i)} \tag{7}$$

$$J(t) = J_g + \sum_{j=1}^n J_j \left(1 - e^{(-t/\tau_j)}\right) \tag{8}$$

where $E_e$ is the equilibrium modulus; $E_i$ is the relaxation strength; $\rho_i$ is the relaxation time; $J_g$ is the glassy compliance; $J_j$ is the retardation strength; $\tau_j$ is the retardation time; and $t$ is the test time.the expressions of $E'(\omega)$, $E''(\omega)$, $J'(\omega)$ and $J''(\omega)$ can be obtained by using Carson transform as [28–31]:

$$E'(\omega) = E_e + \sum_{i=1}^m E_i \frac{\omega^2 \rho_i^2}{\omega^2 \rho_i^2 + 1} \tag{9}$$

$$E''(\omega) = \sum_{i=1}^m E_i \frac{\omega \rho_i}{\omega^2 \rho_i^2 + 1} \tag{10}$$

$$J'(\omega) = J_g + \sum_{j=1}^n J_j \frac{1}{\omega^2 \tau_j^2 + 1} \tag{11}$$

$$J''(\omega) = \sum_{j=1}^n J_j \frac{\omega \tau_j}{\omega^2 \tau_j^2 + 1} \tag{12}$$

From Eqs (9) and (10), it can be known that under the specified relaxation time, $E_e$ and $E_i$ can be obtained by fitting the $E'(\omega)$ and $E''(\omega)$ at different frequencies. The obtained parameters are put into Eq (7) to get the master curve of $J(t)$.

**3.4.2 Determining the discrete retardation spectrum models.**   $E(t)$ and $J(t)$ satisfy the integral relationship shown in Eq (6), and the Prony series expressions of $E(t)$ and $J(t)$ are brought into this equation, and the discrete retardation spectrum can be obtained. Substituting

Eqs (7) and (8) into Eq (14), we get

$$A_{kj}J_j = B_k \tag{13}$$

Where

$$A_{kj} = \begin{cases} E_e\left(1 - e^{-\left(t_k/\tau_j\right)}\right) + \sum_{i=1}^{m} \frac{\rho_i E_i}{\rho_i - \tau_j}\left(e^{-(t_k/\rho_i)} - e^{-\left(t_k/\tau_j\right)}\right) & \text{when } \rho_i \neq \tau_j \\ or \\ E_e\left(1 - e^{-\left(t_k/\tau_j\right)}\right) + \sum_{i=1}^{m} \frac{\rho_i E_i}{\tau_j} e^{-(t_k/\rho_i)} & \text{when } \rho_i = \tau_j \end{cases} \tag{14}$$

$$B_k = 1 - \left(E_e + \sum_{i=1}^{m} E_i e^{-(t_k/\rho_i)}\right) \Big/ \left(E_e + \sum_{i=1}^{m} E_i\right) \tag{15}$$

Where $k = 1,2,\ldots,p$; $j = 1,2,\ldots,n$; $t_k$ is the upper limit of the integral of formula (6).

From Eqs (14) and (15), it can be seen that when the discrete relaxation spectrum and $E_e$ are known $\tau_j$ and $t_k$ are set, the discrete retardation spectrum models can be obtained by solving the inhomogeneous linear equation system, and then $E(t)$ can be obtained.

## 3.5 Determining the continuous spectrum

**3.5.1 Determining the continuous relaxation spectrum models.**   When the time interval of discrete spectrum is close to infinitely small, it will evolve into continuous spectrum. The conversion between the models of the viscoelastic parameters and the continuous spectrum can be realized by using the integral transformation theory. According to the internal relationship between the storage modulus model (ie the GSM) and the continuous relaxation spectrum models $H(\rho)$, the functional relationship of $H(\rho)$ on the model of $E'(\omega)$ can be deduced.

The integral relationship between $E(t)$ and the $H(\rho)$ is [32]

$$E(t) = E_e + \int_{-\infty}^{+\infty} H(\rho)e^{-(t/\rho)}\,\mathrm{dln}\rho \tag{16}$$

According to the Eq (18), the integral expression of the $E^*(\omega)$ with respect to $H(\rho)$ is obtained by calculating:

$$E^*(\omega) = E_e + \int_{-\infty}^{+\infty} H(\rho)\frac{\omega^2\rho^2 + i\omega\rho}{\omega^2\rho^2 + 1}\,\mathrm{dln}\rho \tag{17}$$

Separating the real and imaginary parts of Eq (17), the integral relation of the $E'(\omega)$ and $E''(\omega)$ with respect to the time spectrum is obtained as

$$E'(\omega) = E_e + \int_{-\infty}^{+\infty} H(\rho)\frac{\omega^2\rho^2}{\omega^2\rho^2 + 1}\,\mathrm{dln}\rho \tag{18}$$

$$E''(\omega) = \int_{-\infty}^{+\infty} H(\rho)\frac{\omega\rho}{\omega^2\rho^2 + 1}\,\mathrm{dln}\rho \tag{19}$$

Let $F(\rho) = X^2(\rho) + Y^2(\rho)$ and $G = \arctan Y(\rho)/X(\rho)$, use Euler's formula again to further simplify the formula, and finally obtain the relationship between $H(\rho)$ and the parameters of the

model of $E'(\omega)$ as

$$H(\rho) = -\frac{2}{\pi}\exp\left(A + F^{\frac{1}{2\lambda}}\cos\frac{G}{\lambda}\right)\sin\left(F^{\frac{1}{2\lambda}}\sin\frac{G}{\lambda}\right) \tag{20}$$

**3.5.2 Determining the continuous retardation spectrum.** As shown in Eq (21), the retardation spectrum function can be derived from the known relaxation spectrum function [33]. In order to ensure a one-to-one correspondence between the retardation time and the relaxation time during calculation, the retardation time is also represented by $\tau$.

$$L(\tau) = \frac{H(\tau)}{Z^2(\tau) + \pi^2 H^2(\tau)} \tag{21}$$

$$Z(\tau) = E_e + \int_0^{+\infty}\frac{H(u)}{u - \tau}du \tag{22}$$

# 4 Results and discussion

## 4.1 Linear viscoelasticity characterization of basalt fiber asphalt mixture

**4.1.1 Master curves of $|E^*(\omega)|$ and $\varphi(\omega)$.** Use Excel's Solver function to obtain the model parameters of $|E^*(\omega)|$ and $\varphi(\omega)$. The fitting results are shown in Table 1, and the master curves of $|E^*(\omega)|$ and $\varphi(\omega)$ are shown in Figs 2 and 3.

To compare the effects of basalt fiber content on the master curve of $|E^*(\omega)|$ and $\varphi(\omega)$, the two master curves of viscoelastic parameters are constructed in Fig 4(A) and 4(B). As shown in Fig 4(A), at high frequency, the dynamic modulus of BF-2 is the smallest, the dynamic modulus of BF-0 is the largest, and the dynamic modulus of BF-1 and BF-3 are both smaller than BF-0. At low frequency, the dynamic modulus of BF-1, BF-2 and BF-3 are all larger than BF-0, and the dynamic modulus of the three groups of basalt fiber asphalt mixtures have little difference. According to TTPS, it shows that the basalt fiber asphalt has good mixed deformation ability and strong toughness at low-temperature, and can better resist the action of external loads and disperse the stress.

As shown in Fig 4(B), under high and low frequency conditions, the phase angles of BF-1, BF-2and BF-3 are lower than BF-0; among them, BF-2 has the smallest phase angle. This is because the basalt fibers adsorb the asphalt, which reduces the viscoelasticity ratio of the asphalt mixture, and the fibers form a grid structure in the asphalt mixture, which constrains the deformation of the asphalt mixture and reduces the phase angle. This is because basalt fibers improves the low-temperature deformation capacity of asphalt mixtures by changing their the internal structure.

**Table 1. Models parameters of $|E^*(\omega)|$ and $\varphi(\omega)$.**

| Types of asphalt mixtures | Parameters | | | | | | | Error (%) |
|---|---|---|---|---|---|---|---|---|
| | $\delta$ | $\alpha$ | $\lambda$ | $\beta$ | $\gamma$ | $C_1$ | $C_2$ | |
| BF-0 | 1.95 | 2.77 | 0.76 | 0.02 | -0.48 | 15.85 | 200.43 | 5.77 |
| BF-1 | 2.21 | 2.41 | 0.84 | -0.03 | -0.64 | 23.93 | 325.36 | 5.46 |
| BF-2 | 2.36 | 2.13 | 0.83 | 0.04 | -0.70 | 24.27 | 351.58 | 5.20 |
| BF-3 | 2.08 | 2.57 | 0.88 | -0.11 | -0.58 | 28.48 | 350.00 | 4.25 |

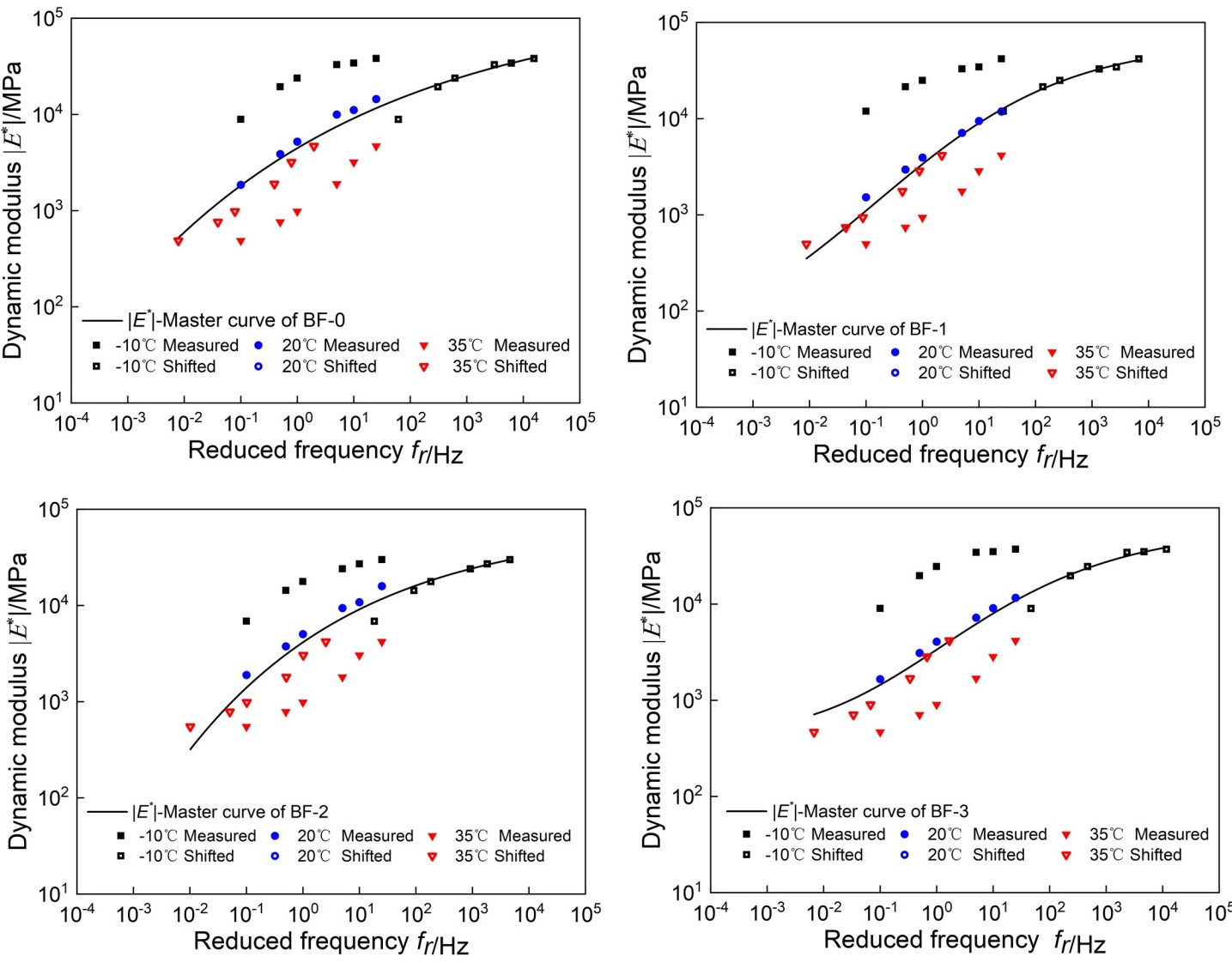

**Fig 2. Master curves of $|E^*(\omega)|$.**

**4.1.2 Master curves of $E'(\omega)$ and $E''(\omega)$.** The model parameters of $E'(\omega)$ and $E''(\omega)$ are simultaneously obtained by minimizing the parameter error. The fitting results are shown in Table 2, and the master curves of $E'(\omega)$ and $E''(\omega)$ are shown in Figs 5 and 6.

It can be seen from Fig 7 that at high frequencies, the storage modulus first decreases and then increases with the increase of basalt fiber content, and the loss modulus decreases with the increase of basalt fiber content. It shows that at low temperature, adding basalt fiber can reduce the energy stored and lost in the asphalt mixture, and correspondingly reduce its low-temperature elastic capacity and low-temperature viscous deformation. At low frequency, both storage modulus and loss modulus increase then decrease then increase with the increase of basalt fiber content.

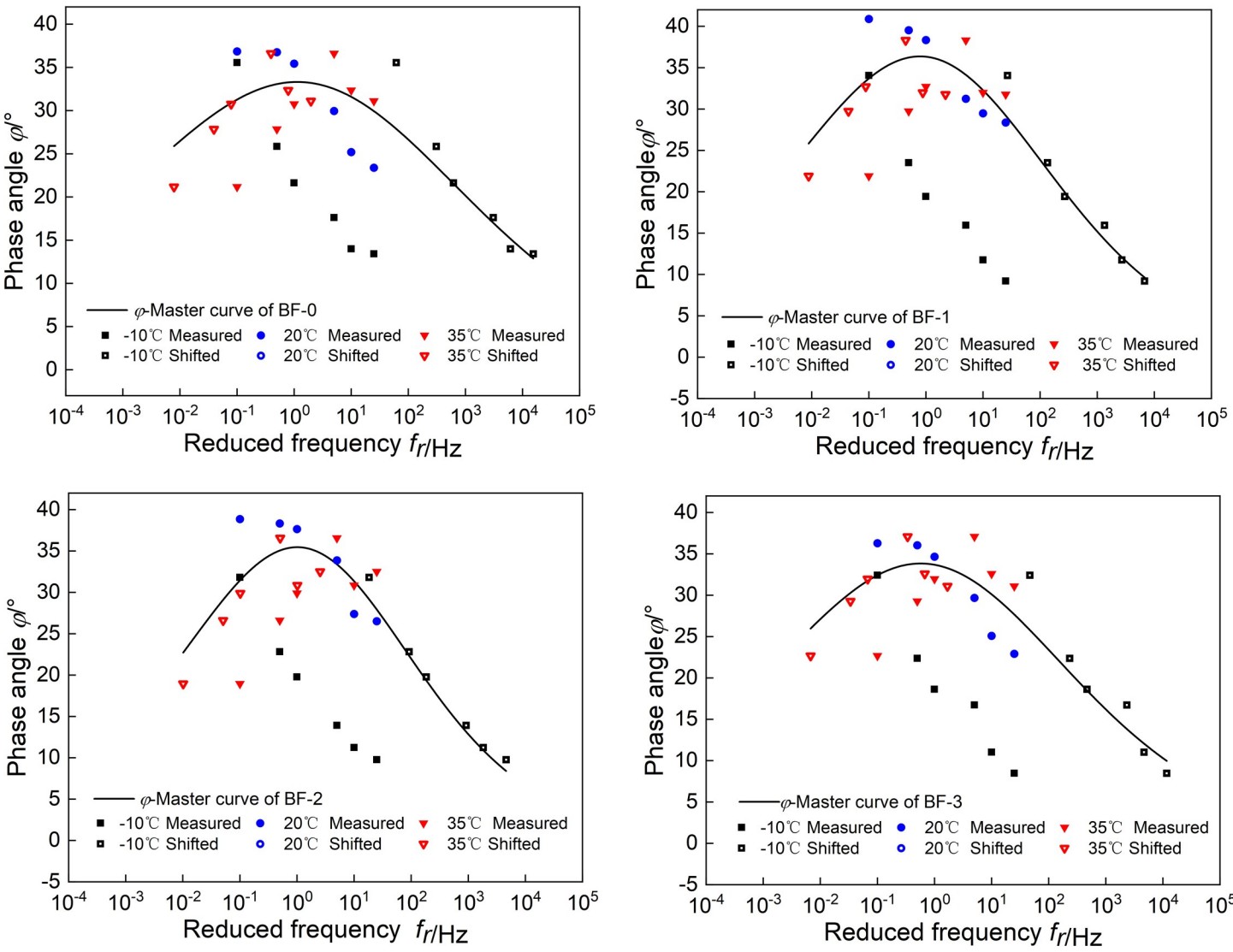

**Fig 3. Master curves of $\varphi(\omega)$.**

## 4.2 Construction of master curves of *E(t)* and *J(t)* from discrete spectrum

**4.2.1 Determining the discrete relaxation spectrum.** A discrete relaxation spectrum containing all viscoelastic information was obtained using or the collocation method. When calculating according to Eqs (9) and (10), the number of Maxwell units needs to be determined; in general, the more Maxwell units the more accurate the results are, but too many units will increase the difficulty of fitting. Therefore, 15 Maxwell units are selected for the calculation in order to ensure the accuracy of the results and to reduce the difficulty of fitting. The Prony series coefficients for the relaxation modulus can be obtained by minimizing the target error, as shown in Table 3.

**4.2.2 Derivation of discrete retardation spectrum from discrete relaxation spectrum.** The discrete retardation spectrum models can be derived. This process generally requires three steps, namely determining the retardation time $t_k$, calculating the retardation intensity $Jj$ and determining the glassy compliance $J_g$. Firstly, the horizontal coordinate of the graph

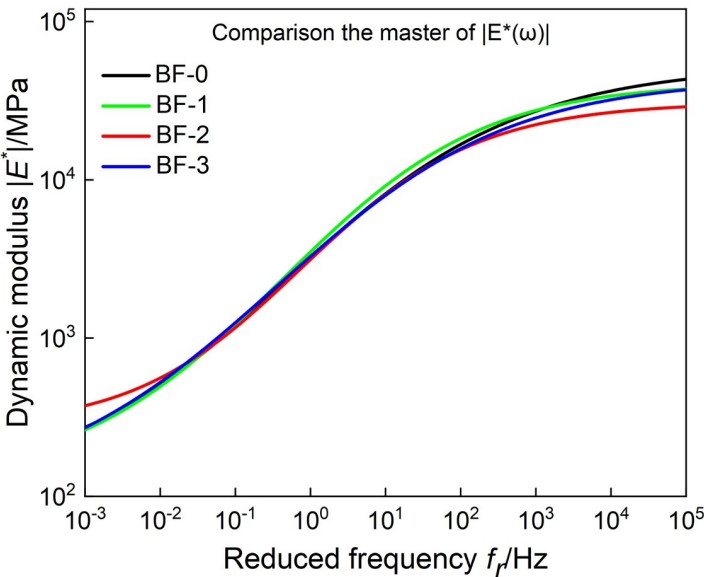
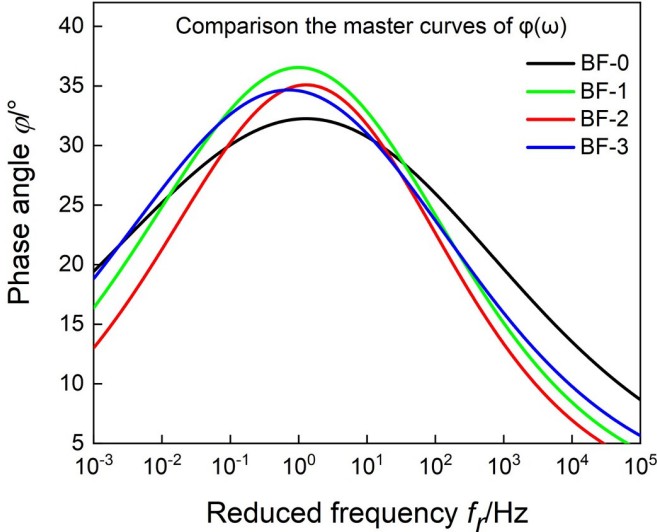

**Fig 4. Comparison the master curves of $|E^*(\omega)|$ and $\varphi(\omega)$.**

minimum obtained by the graphical rooting method is the retardation time $t_k$, and then the error function is solved to determine the retardation intensity $J_j$, and finally, the glassy compliance $J_g$ is determined according to the Laplace's initial and final theorems. Table 4 shows the coefficients of the Prony series of creep softness obtained from the conversion of the relaxation time spectra of the four asphalt mixtures [34].

The discrete relaxation and retardation spectrum are drawn according to Tables 3 and 4, as shown in Figs 8 and 9. Both discrete spectrum have obvious asymmetry. And with the increase of basalt fiber content, the peak relaxation strength showed a trend of decreasing first, while the discrete retardation spectrum did not have a specific change rule.

**4.2.3 Construction the master curves of *E(t)* and *J(t)* from discrete spectrum.** The Prony series parameter of relaxation modulus in Table 3 is put into Eq (7), and the mater curve of relaxation modulus can be obtained by conversion, as shown in Fig 10. And by putting the creep compliance Prony series parameters in Table 4 into Eq (8), and the relaxation modulus master curve of creep compliance can be constructed, as shown in Fig 11.

As shown in Fig 10, the relaxation modulus reaches its maximum value when the loading time is short. When the loading time is very long, the relaxation modulus reaches the minimum value. And the relaxation modulus was found to vary greatly from $10^{-5}$ to $10^{5}$. This is because, the internal stress of the asphalt mixture is accumulated for a short loading time. With the extension of the loading time, the internal stress dissipates rapidly, and a small amount of stress remains after a long time of relaxation. At the same time, in a short loading

**Table 2. Models parameters of *E'(ω)* and *E''(ω)*.**

| Types of asphalt mixtures | Parameters | | | | | | | Error (%) |
|---|---|---|---|---|---|---|---|---|
| | $\delta'$ | $\alpha'$ | $\lambda$ | $\beta'$ | $\gamma'$ | $C_1'$ | $C_2'$ | |
| BF-0 | 2.17 | 2.70 | 0.88 | 0.07 | -0.60 | 20.72 | 274.93 | 1.03 |
| BF-1 | 2.37 | 2.39 | 0.33 | -0.11 | -0.62 | 28.30 | 381.81 | 1.39 |
| BF-2 | 2.50 | 2.13 | 0.33 | -0.03 | -0.65 | 26.59 | 382.06 | 1.70 |
| BF-3 | 2.29 | 2.48 | 0.24 | -0.14 | -0.53 | 19.89 | 250.94 | 1.14 |

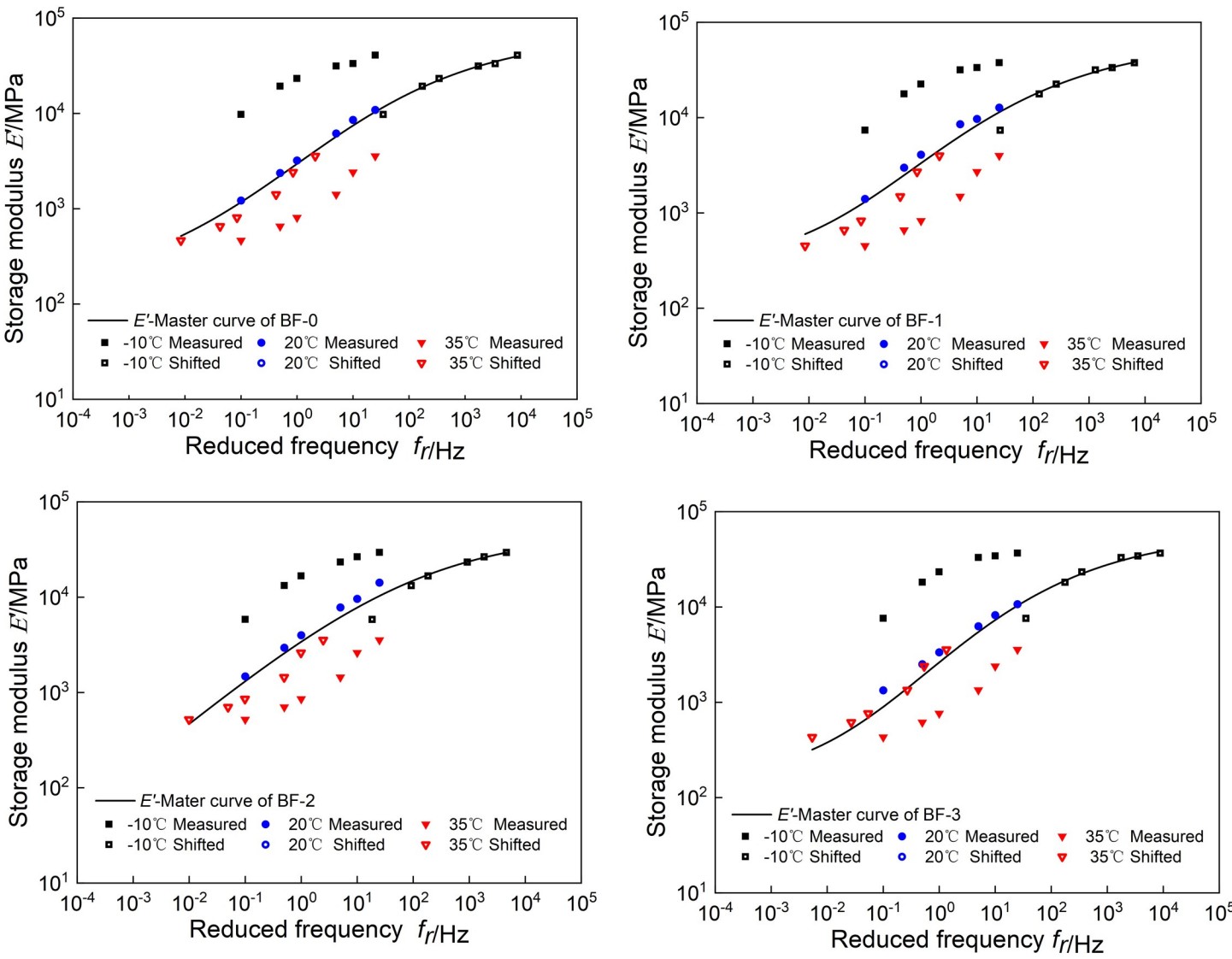

**Fig 5. Master curves of E'(ω).**

time, the relaxation modulus decreased with the increase of basalt fiber content and then increased, and the relaxation modulus was the smallest when the basalt fiber content was 0.2%. It shows that basalt fiber can improve the low-temperature relaxation ability and the thermal shrinkage cracking resistance of asphalt mixtures, and reduce the accumulation of temperature stress of asphalt mixture. This is because basalt fiber improves the stress diffusion ability of the asphalt mixture, thereby improving the low-temperature crack resistance. In a long loading time, the relaxation modulus first increased and then decreased with the increase of basalt fiber content, and the relaxation modulus reached the maximum when the basalt fiber content was 0.2%.

As can be seen in Fig 11, when the loading time is short, the creep compliance tends to the minimum value; when the loading time is very long, the creep compliance tends to the maximum value, and the creep compliance changes the largest from $10^{-5}$ to $10^5$s. This phenomenon can be attributed to the softening of asphalt at high-temperature. As a whole, the asphalt

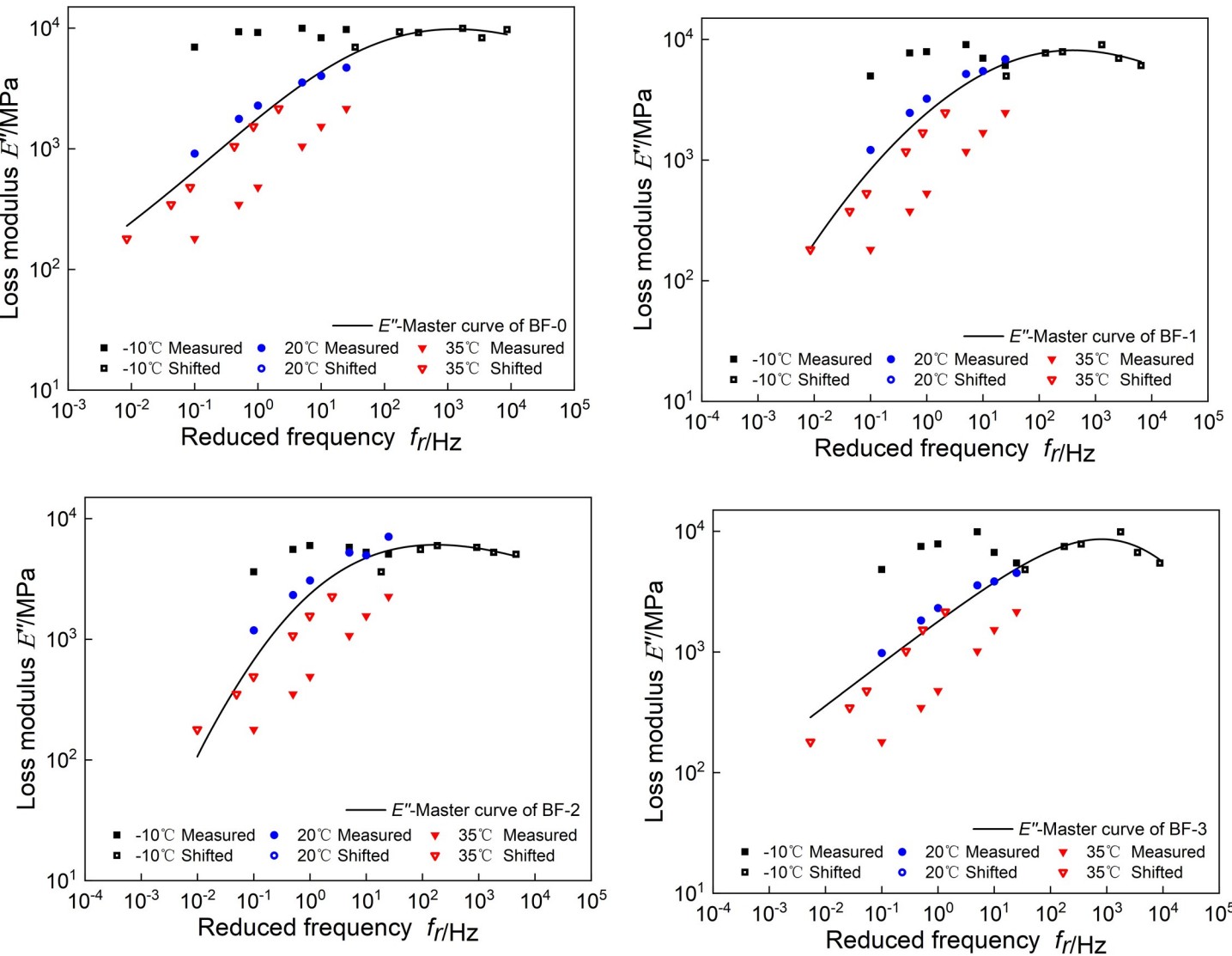

**Fig 6. Master curves of $E''(\omega)$.**

mixture is composed of additives such as coarse and fine aggregates, asphalt, and mineral powder. With the increase of temperature or loading time, the viscosity of asphalt increases. The effect of the knot is weakened, which leads to a reduction in the overall resistance to deformation of the asphalt. At the same time, in the short loading time, the creep compliance first increases and then decreases with the basalt fiber content, and the creep compliance of BF-2 is the largest. For a longer loading time, the creep compliance decreases and then increases with the increase of the basalt fiber content, and the creep compliance of BF-2 is the smallest. This shows that basalt fiber can improve the deformation ability and resist creep deformation at high temperature of the asphalt mixture. And correspondingly improve its anti-rutting ability. Moreover, when the basalt fiber content is 0.2%, the effect of improving the deformation resistance of the asphalt mixture is the best.

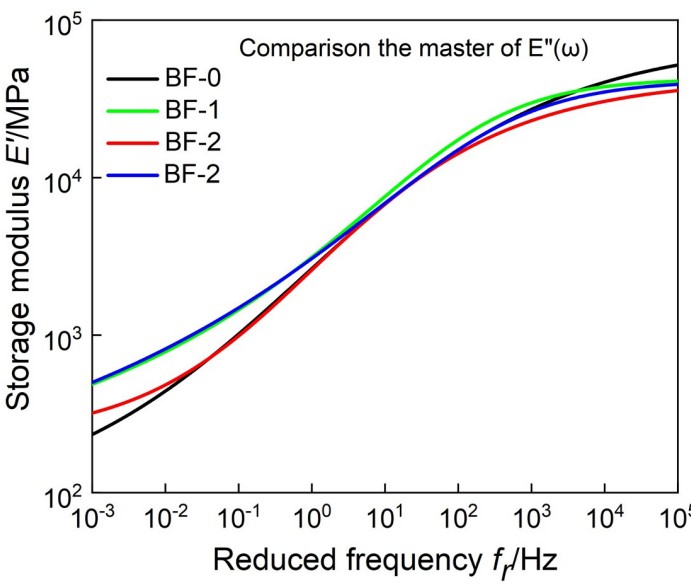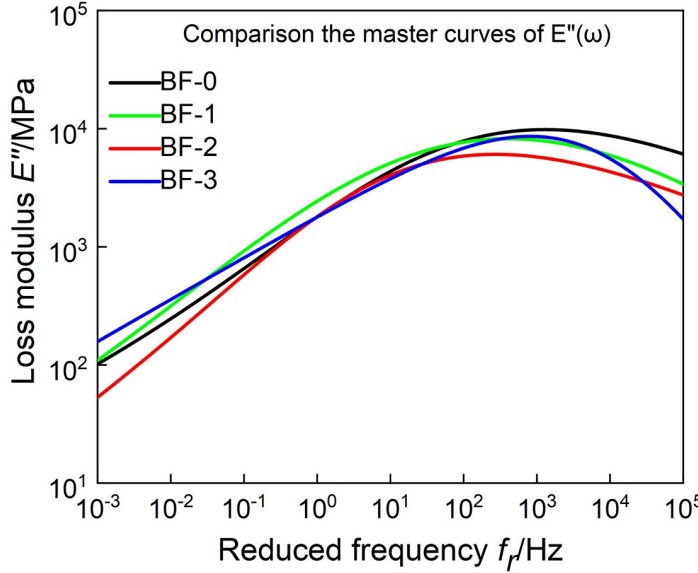

**Fig 7. Comparison the master curves of $E'(\omega)$ and $E''(\omega)$.**

## 4.3 Construction master curves of $E(t)$ and $J(t)$ from continuous spectrum

**4.3.1 Determining the continuous relaxation spectrum.** The model parameters of $E'(t)$ and $E''(t)$ in Table 3 are put into Eq (20), and the continuous relaxation spectrum model of four kinds of asphalt mixture can be obtained. Taking the relaxation time as the abscissa and $H(\rho)$ as the ordinate, the continuous relaxation spectrum when the reference temperature is 20˚C is obtained, as shown in Fig 12. The relaxation spectrum is asymmetric. And the peak of the continuous spectrum is the same as that obtained by the discrete relaxation time spectrum. The parametric instruments used were derived from master curves of $E'(t)$ and $E''(t)$. The

**Table 3. Prony series coefficients of $E(t)$.**

| $i$ | $\rho_i$ | $E_i$ | | | |
|---|---|---|---|---|---|
| | | **BF-0** | **BF-1** | **BF-2** | **BF-3** |
| 1 | 2.00E-07 | 2003.47 | 2007.60 | 1990.01 | 1752.87 |
| 2 | 2.00E-06 | 3288.98 | 3270.51 | 3094.23 | 2869.47 |
| 3 | 2.00E-05 | 10362.20 | 8848.07 | 7263.62 | 10359.91 |
| 4 | 2.00E-04 | 14549.99 | 12328.44 | 8601.48 | 11342.64 |
| 5 | 2.00E-03 | 10263.25 | 11284.45 | 9341.45 | 9963.79 |
| 6 | 2.00E-02 | 5496.59 | 6596.66 | 5678.70 | 5287.52 |
| 7 | 2.00E-01 | 2170.16 | 2265.82 | 2130.64 | 1958.89 |
| 8 | 2.00E+00 | 679.75 | 616.08 | 512.75 | 645.95 |
| 9 | 2.00E+01 | 214.88 | 144.69 | 138.20 | 161.07 |
| 10 | 2.00E+02 | 90.52 | 88.04 | 36.66 | 81.63 |
| 11 | 2.00E+03 | 50.19 | 40.32 | 34.56 | 40.94 |
| 12 | 2.00E+04 | 54.32 | 42.84 | 42.33 | 25.84 |
| 13 | 2.00E+05 | 34.86 | 12.57 | 18.61 | 21.62 |
| 14 | 2.00E+06 | 13.34 | 2.95 | 9.05 | 11.62 |
| 15 | 2.00E+07 | 13.34 | 2.95 | 9.05 | 11.62 |
| $E_e$ | | 161.00 | 218.33 | 312.02 | 176.15 |

**Table 4. Prony series coefficients of $J(t)$.**

| $j$ | BF-0 | | BF-1 | | BF-2 | | BF-3 | |
|---|---|---|---|---|---|---|---|---|
| | $\tau_j$ | $J_j$ | $\tau_j$ | $J_j$ | $\tau_j$ | $J_j$ | $\tau_j$ | $J_j$ |
| 1 | 2.11E-07 | 7.09E-07 | 2.11E-07 | 6.59E-07 | 2.11E-07 | 2.36E-07 | 2.11E-07 | 1.03E-06 |
| 2 | 2.18E-06 | 2.05E-06 | 2.18E-06 | 2.26E-06 | 2.18E-06 | 4.44E-06 | 2.18E-06 | 8.82E-07 |
| 3 | 2.60E-05 | 5.40E-06 | 2.48E-05 | 4.35E-06 | 2.54E-05 | 6.17E-06 | 2.60E-05 | 1.04E-05 |
| 4 | 3.44E-04 | 1.74E-05 | 3.12E-04 | 1.71E-05 | 2.89E-04 | 1.60E-05 | 3.04E-04 | 1.44E-05 |
| 5 | 4.12E-03 | 5.48E-05 | 4.02E-03 | 3.76E-05 | 3.92E-03 | 4.46E-05 | 3.82E-03 | 4.12E-05 |
| 6 | 4.93E-02 | 1.14E-04 | 5.44E-02 | 1.29E-04 | 5.17E-02 | 1.38E-04 | 4.93E-02 | 1.16E-04 |
| 7 | 5.34E-01 | 3.75E-04 | 5.90E-01 | 4.21E-04 | 5.90E-01 | 4.58E-04 | 5.21E-01 | 3.25E-04 |
| 8 | 4.42E+00 | 8.80E-04 | 4.65E+00 | 1.09E-03 | 4.11E+00 | 8.67E-04 | 5.00E+00 | 7.89E-04 |
| 9 | 3.16E+01 | 1.03E-03 | 2.79E+01 | 9.55E-04 | 2.66E+01 | 5.99E-04 | 4.24E+01 | 1.92E-03 |
| 10 | 2.62E+02 | 8.53E-04 | 2.55E+02 | 7.40E-04 | 2.20E+02 | 3.17E-04 | 3.78E+02 | 2.67E-03 |
| 11 | 2.39E+03 | 6.70E-04 | 2.27E+03 | 4.59E-04 | 2.17E+03 | 2.06E-04 | 2.77E+03 | 2.69E-03 |
| 12 | 2.47E+04 | 5.48E-04 | 2.35E+04 | 2.05E-04 | 2.24E+04 | 1.55E-04 | 3.23E+04 | 4.18E-03 |
| 13 | 2.37E+05 | 3.01E-04 | 2.15E+05 | 1.56E-04 | 2.15E+05 | 1.06E-04 | 2.43E+05 | 2.88E-03 |
| 14 | 2.16E+06 | 1.86E-04 | 2.06E+06 | 1.02E-04 | 2.06E+06 | 8.60E-05 | 2.06E+06 | 2.33E-03 |
| 15 | 2.18E+07 | 2.70E-05 | 2.02E+07 | 2.61E-05 | 2.07E+07 | 9.22E-05 | 2.02E+07 | 3.04E-04 |
| $J_g$ | 2.02E-05 | | 2.09E-05 | | 2.55E-05 | | 2.24E-05 | |

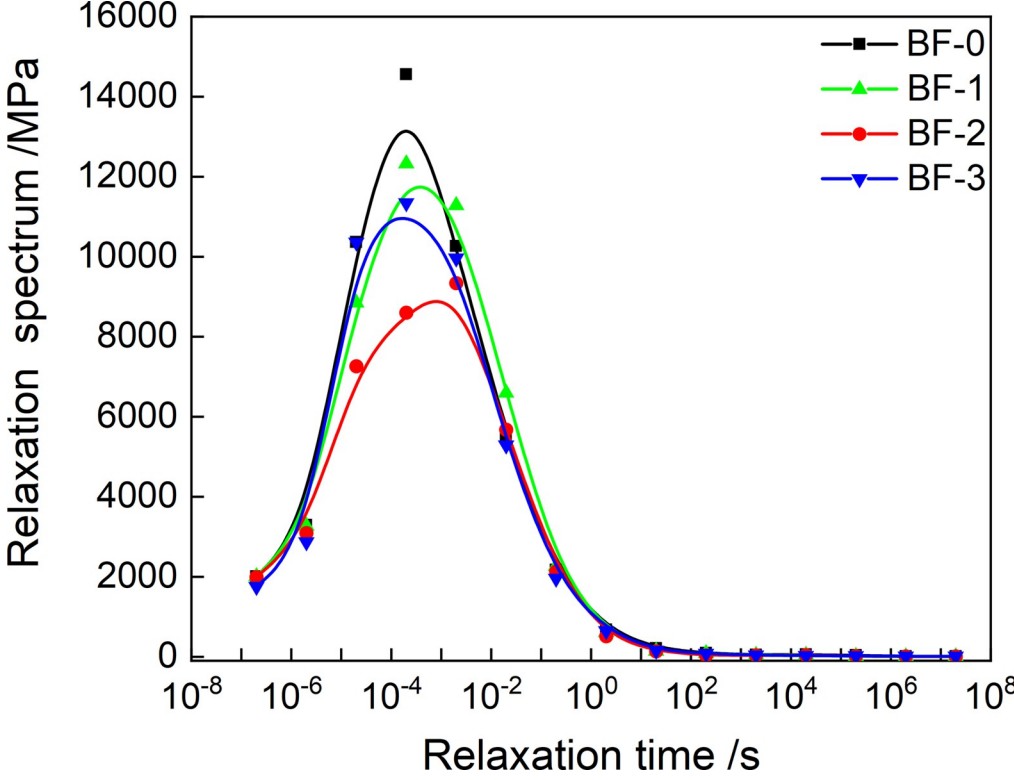

**Fig 8. Discrete relaxation spectrum.**

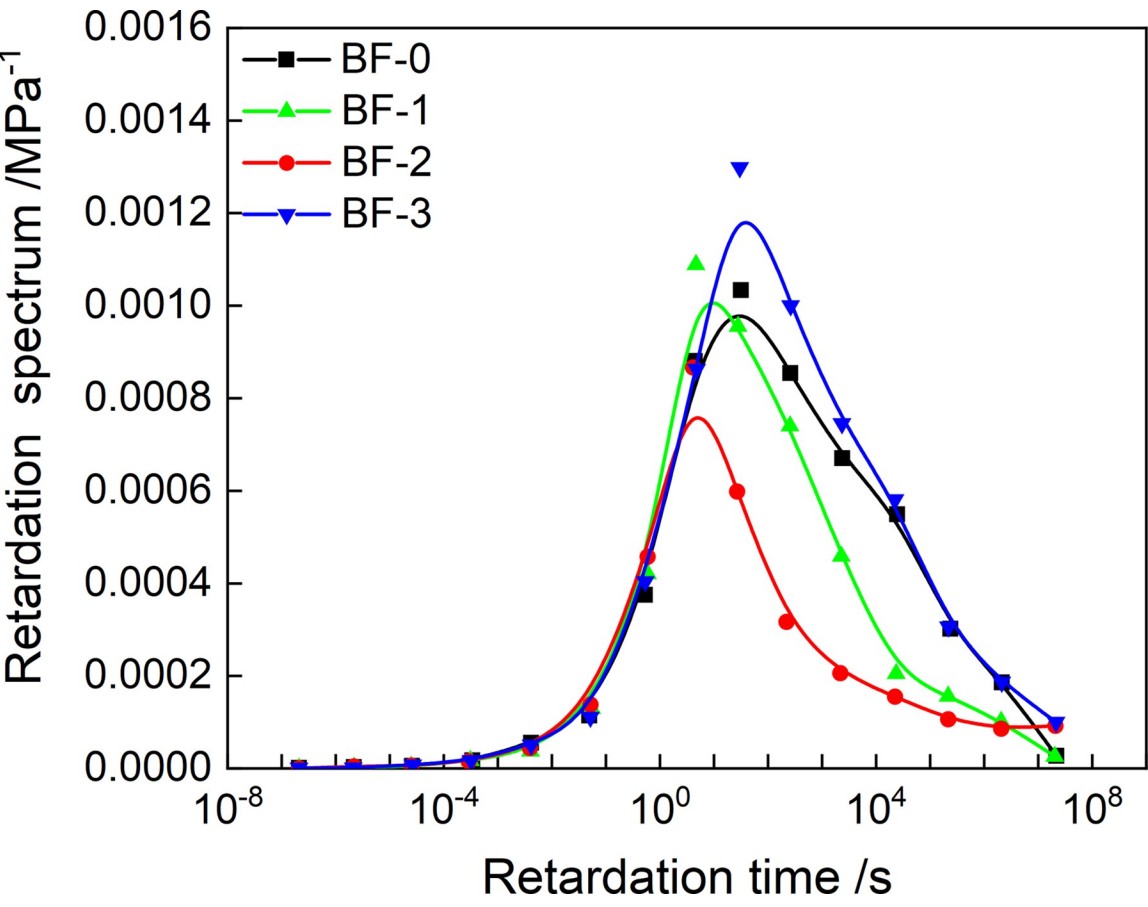

**Fig 9. Discrete retardation spectrum.**

constructed continuous relaxation spectrum therefore contains all the linear viscoelastic information.

**4.3.2 Constructed the continuous retardation spectrum from the continuous relaxation spectrum.** When deriving the continuous retardation spectrum from Eq (21), the integral term $Z(\tau)$ and the equilibrium modulus $E_e$ need to be calculated. Use the chained trapezoidal rule to solve the integral term $Z(\tau)$ .The integration interval of $\tau$ is selected as $[10^{-40},10^{30}]$, the integration interval of $u$ is selected as $[10^{-40},10^{30}]$, the number of sub-intervals is 14000, and the length of the sub-interval is $\log\Delta\tau = \log\Delta u = 0.05"$. The equilibrium modulus $E_e$ can be calculated from Eq (23).

$$E_e = E'(\omega = 0) = E'(f_r = 0) = 10^{\delta} \tag{23}$$

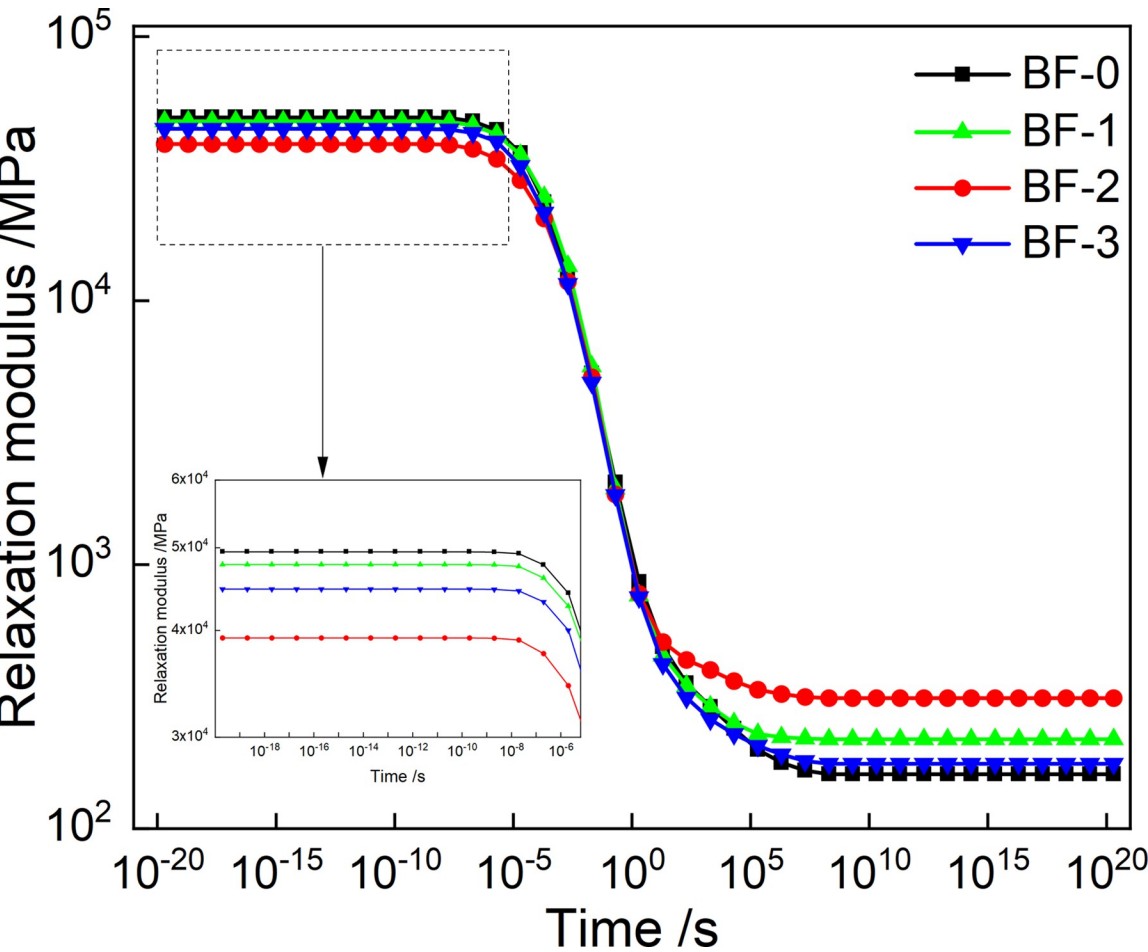

**Fig 10. Master curve of $E(t)$ from discrete relaxation spectrum.**

The numerical expression of $Z(\tau)$ is

$$Z(\tau) = 10^{\delta} + (\ln 10) \cdot \frac{0.005}{2} \cdot$$

$$\left\{ \begin{array}{l} \dfrac{10^{-40.0025}}{10^{-40.0025} - \tau} H\left(10^{-40.0025}\right) \\[2ex] +2\left[\dfrac{10^{-40.0025}}{10^{-40.0025} - \tau} H\left(10^{-40.0025}\right) + \cdots + \dfrac{10^{29.9925}}{10^{29.9925} - \tau} H\left(10^{29.9925}\right)\right] \\[2ex] + \dfrac{10^{29.9975}}{10^{29.9975} - \tau} H\left(10^{29.9975}\right) \end{array} \right\} \tag{24}$$

Substituting the discretized values of $Z(\tau)$ and $H(\tau)$ into formula (21) can obtain the continuous retardation spectrum values of the four asphalt mixtures, and generate the continuous retardation spectrum shown in Fig 13. As shown in Fig 14, the variation law of the peak retardation intensity of the continuous retardation spectrum with the fiber content is the same as that of the discrete retardation spectrum. However, the continuous retardation spectrum has lower peak retardation intensity than the discrete retardation spectrum.

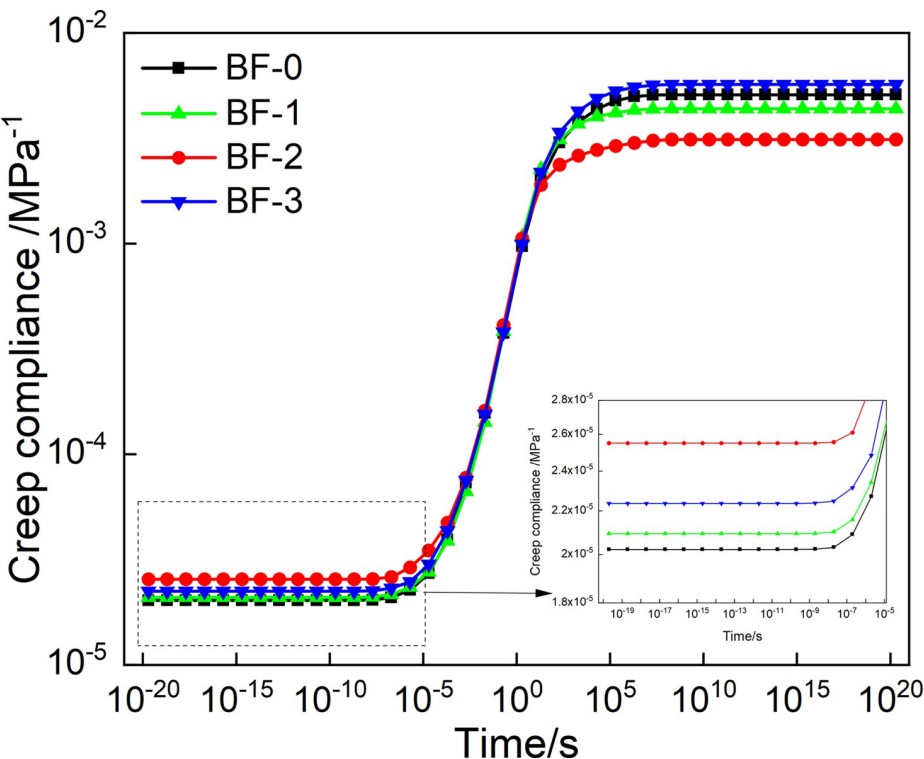

**Fig 11. Master curve of _J(t)_ from discrete retardation spectrum.**

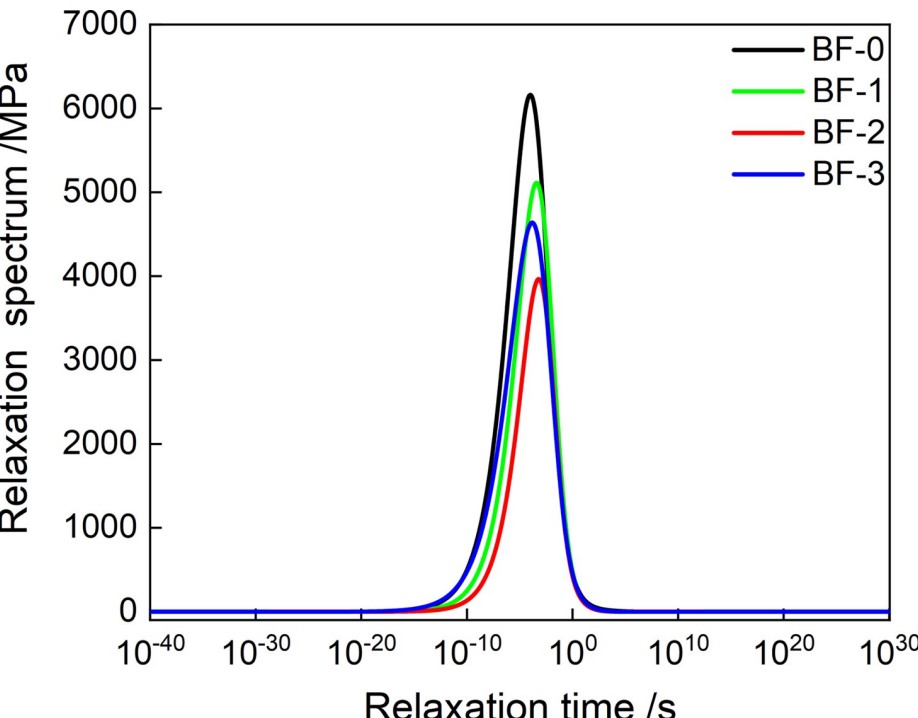

**Fig 12. Continuous relaxation spectrum of asphalt mixtures.**

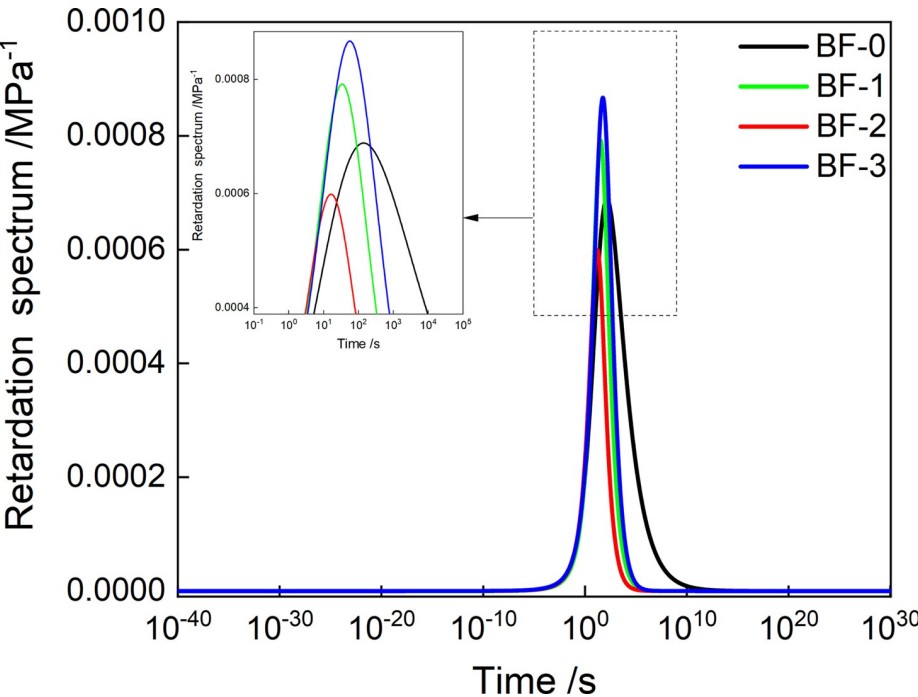

**Fig 13. Continuous retardation spectrum of asphalt mixtures.**

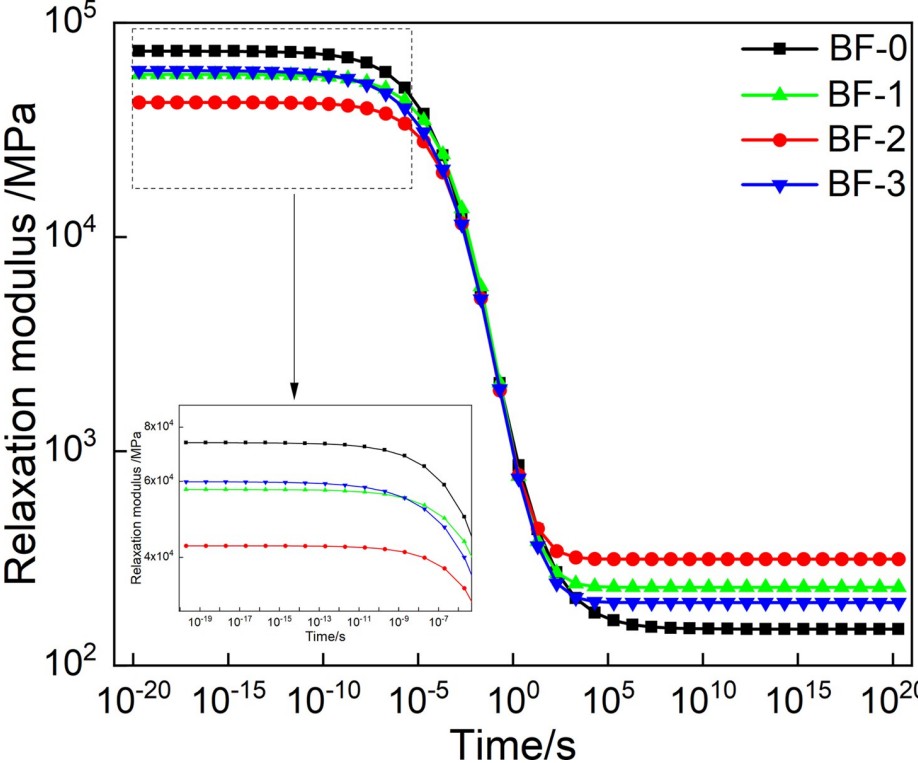

**Fig 14. Master curve of *E(t)* from continuous relaxation spectrum.**

**4.3.3 Constructed master curves of *E(t)* and *J(t)* based on continuous spectrum.** Use the chained trapezoidal rule to solve Eq (16), the integration interval is selected as $[10^{-40}, 10^{30}]$, and the subinterval length is selected as $\log\Delta\rho = 0.005$. The numerical expression of *E(t)* is shown in Eq (25). The models of *E(t)* of the four asphalt mixtures were calculated according to Eq (25), and construct the master curves, as shown in Fig 14. The master curves of *E(t)* obtained from continuous and discrete relaxation spectrum have the same trend of variation with loading time. Moreover, at short and long loading times, the master curves of *E(t)* with the basalt fiber content has the same trend, also.

$$E(t) \approx E_e + (\ln 10) \cdot \frac{0.005}{2} \cdot$$
$$\begin{Bmatrix} H(10^{-40})e^{-(t/10^{-40})} \\ +2\left[H(10^{-39.995})e^{-(t/10^{-39.995})} + \cdots + H(10^{29.995})e^{-(t/10^{29.995})}\right] \\ +H(10^{30})e^{-(t/10^{30})} \end{Bmatrix} \quad (25)$$

The model of *J(t)* can be calculated from Eq (26). Calculate Eq (26) using the same method and settings as for relaxation modulus. The numerical expression of *J(t)* with respect to $L(\tau)$ is shown in Eq (27).

$$J(t) = J_g + \int_{-\infty}^{+\infty} L(\tau)\left(1 - e^{-(t/\tau)}\right)\mathrm{dln}\tau \quad (26)$$

$$J(t) \approx J_g + (\ln 10) \cdot \frac{0.005}{2} \cdot$$
$$\begin{Bmatrix} L(10^{-40})\left(1 - e^{-(t/10^{-40})}\right) \\ +2\left[L(10^{-39.995})\left(1 - e^{-(t/10^{-39.995})}\right) + \cdots + L(10^{29.995})\left(1 - e^{-(t/10^{29.995})}\right)\right] \\ +L(10^{30})\left(1 - e^{-(t/10^{30})}\right) \end{Bmatrix} \quad (27)$$

In Eq (28), there is still one unknown $J_g$, and according to the initial-value and final-value theorem of Laplace transform $J_g$ and $E'(\omega)$ have the relationship shown in Eq (28).

$$J_g = J(0) = \frac{1}{E(0)} = \frac{1}{E'(\omega \to \infty)} = \frac{1}{10^{\delta+\alpha}} \quad (28)$$

According to Eqs (27) and (28), the master curve of *J(t)* can be construct, as shown in Fig 15. The master curves of *J(t)* obtained from the continuous and discrete retardation spectrum has the same trend with loading time. Moreover, the creep compliance changes with the basalt fiber content in the same trend in short and long loading time, also. However, the trend of creep flexibility with fiber doping obtained from continuous retardation spectrum is more pronounced than that of discrete retardation spectrum for a very short loading time.

## 4.4 Comparison of master curves differences of *E(t)* and *J(t)*

To evaluate the difference between the master curves of *E(t)* and *J(t)* constructed from two types of spectrum models, a plot of relative error versus loading time was established by calculating the relative error of the relaxation modulus and creep flexural. And the relative errors

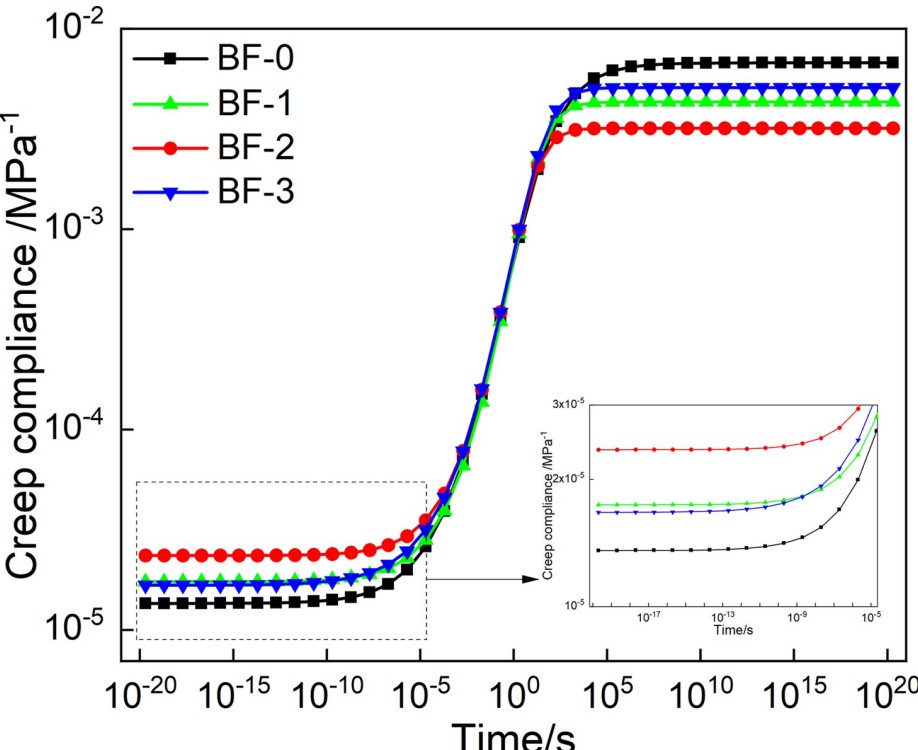

**Fig 15. Master curve of *J(t)* from continuous retardation spectrum.**

are established as a function of Load time change graph, as shown in Figs 16 and 17. In the range $10^{-5}$ to $10^0$s, the errors calculated from two types of spectrum models are the smallest. But significant differences are shown in loading times less than $10^{-5}$s. These reasons can be seen from the continuous and the discrete spectrum.

## 4.5 Accuracy check of the master curves of *E(t)* and *J(t)*

The errors of the master curves of *E(t)* and *J(t)* mainly come from two aspects: on the one hand, the errors generated by using Eq to construct the master curves of *E(ω)* and *E″(ω)*; on the other hand, the error resulting from the transformation of the master curve of the visco-elastic parameters. Therefore, the accuracy of the master curves of *E(t)* and *J(t)* needs to be evaluated.

**4.5.1 Accuracy verification of the master curves of *E(t)*.**    According to Eqs (9) and (10), *E′* and *E″* of each of the four asphalt mixtures at the reduced frequency can be directly calculated.

When calculating *E'(ω)* and *E″(ω)* from *H(ρ)*, Eqs (18) and (19) are calculated using the chained trapezoidal rule, the integral interval is selected as $[10^{-40}, 10^{39}]$, and the sub-interval length is selected as $\log\Delta\rho = 0.05$. The numerical model expressions of *E'(ω)* and *E″(ω)* with

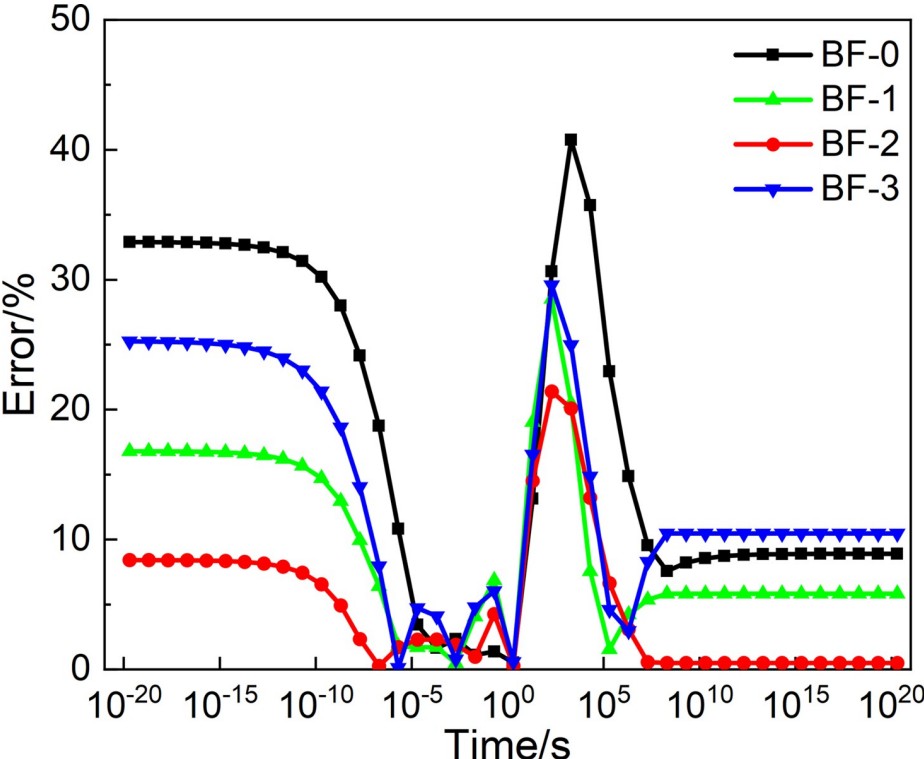

**Fig 16. Relative errors of *E*(*t*).**

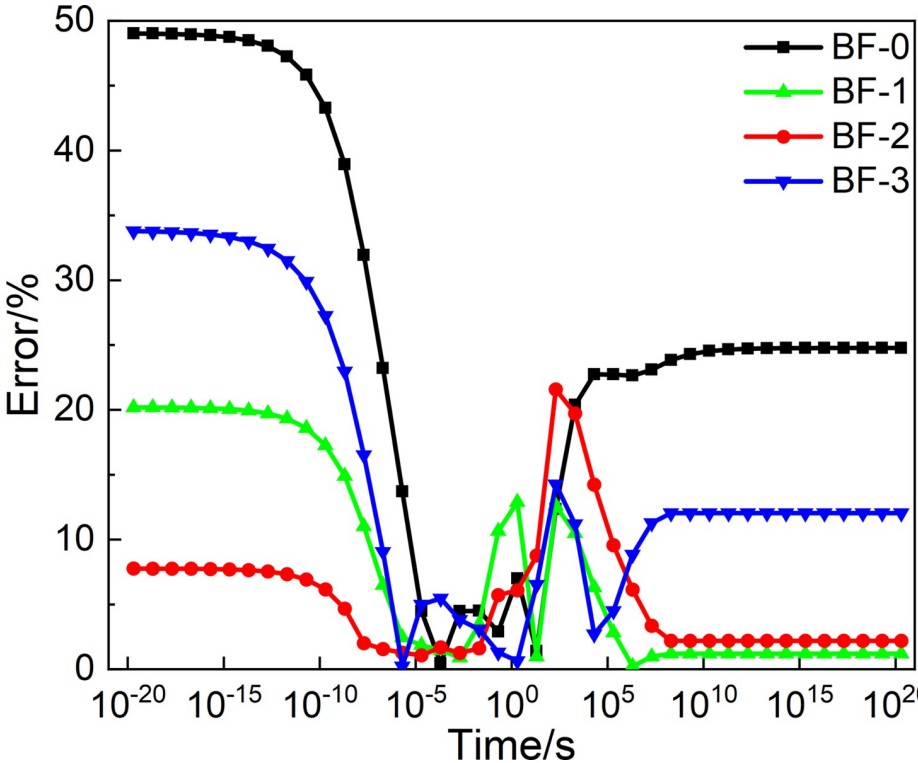

**Fig 17. Relative errors of *J*(*t*).**

respect to $H(\rho)$ are respectively:

$$E'(\omega) \approx 10^{\delta} + (\ln 10) \cdot \frac{0.005}{2} \cdot$$

$$\left\{ \begin{array}{l} H\big(10^{-40}\big) \dfrac{\omega^2 \big(10^{-40}\big)^2}{1 + \omega^2 \big(10^{-40}\big)^2} \\[3mm] + 2\left[ H\big(10^{-39.995}\big) \dfrac{\omega^2 \big(10^{-39.995}\big)^2}{1 + \omega^2 \big(10^{-39.995}\big)^2} + \cdots H\big(10^{29.995}\big) \dfrac{\omega^2 \big(10^{29.995}\big)^2}{1 + \omega^2 \big(10^{29.995}\big)^2} \right] \\[3mm] + H\big(10^{30}\big) \dfrac{\omega^2 \big(10^{30}\big)^2}{1 + \omega^2 \big(10^{30}\big)^2} \end{array} \right\} \quad (29)$$

$$E''(\omega) \approx (\ln 10) \cdot \frac{0.005}{2} \cdot$$

$$\left\{ \begin{array}{l} H\big(10^{-40}\big) \dfrac{\omega 10^{-40}}{1 + \omega^2 \big(10^{-40}\big)^2} + H\big(10^{-39.995}\big) \\[3mm] + 2\left[ H\big(10^{-39.995}\big) \dfrac{\omega 10^{-39.995}}{1 + \omega^2 \big(10^{-39.995}\big)^2} + \cdots + H\big(10^{29.995}\big) \dfrac{\omega 10^{29.995}}{1 + \omega^2 \big(10^{29.995}\big)^2} \right] \\[3mm] H\big(10^{30}\big) \dfrac{\omega 10^{30}}{1 + \omega^2 \big(10^{30}\big)^2} \end{array} \right\} \quad (30)$$

The storage modulus and loss modulus at each reduced frequency are calculated according to Eqs (27) and (28), and their comparison with the master curves is shown in Fig 18. For the four asphalt mixtures, the storage modulus and the master curve almost overlap, but the loss modulus and of the master curve obtained by experiment will have a certain deviation at higher or lower frequencies. And all coefficients of determination $R^2$ are greater than 0.94. The storage modulus or loss modulus calculated by discrete and continuous relaxation spectrum is almost the same as $R^2$. This shows that the discrete and continuous relaxation spectrum models can not only construct the relaxation modulus master curve with high accuracy, but also the obtained relaxation modulus master curve has little difference in accuracy.

**4.5.2 Accuracy verification of the master curves of _J(t)_.**   By the relation that $E^*(\omega)$ and $J^*(\omega)$ are inverses of each other, it is deduced that the storage compliance $J'(\omega)$ and loss compliancein frequency $J''(\omega)$ are related to $E'(\omega)$ and $E''(\omega)$ are as follows:

$$J'(\omega) = \frac{E'(\omega)}{\left[E'(\omega)\right]^2 + \left[E''(\omega)\right]^2} \quad (31)$$

$$J''(\omega) = \frac{E'(\omega)}{\left[E'(\omega)\right]^2 + \left[E''(\omega)\right]^2} \quad (32)$$

$J'(\omega)$ and $J''(\omega)$at reduced frequency are calculated using Eqs (31) and (32). The master curves of $J'(\omega)$ and $J''(\omega)$ are constructed again using the GSM and the approximate K-K relationship.

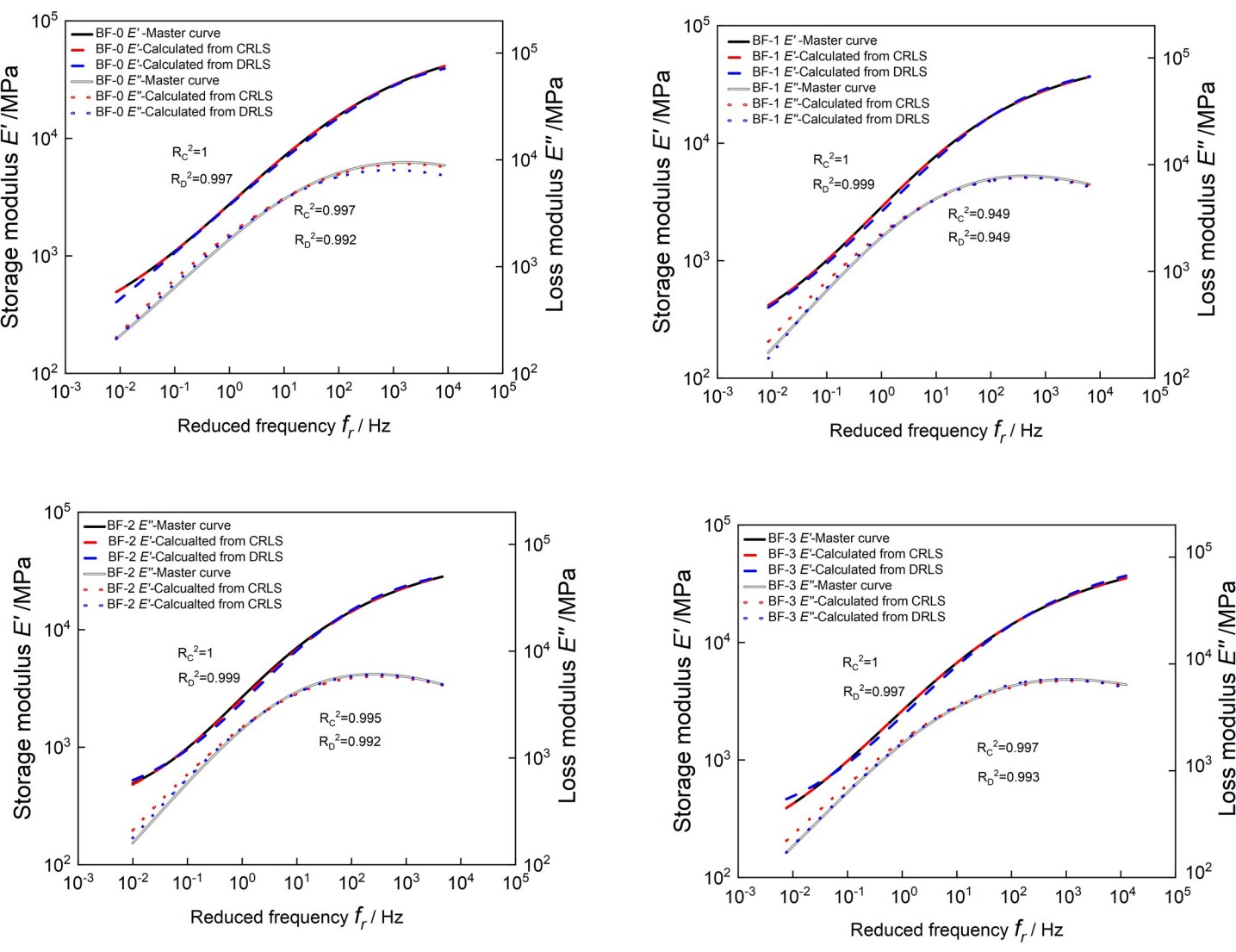

**Fig 18. Comparison of calculated E'(ω) and E''(ω) with the master curves of E'(ω) and E''(ω).**

$J'(\omega)$ and $J''(\omega)$ for each reduced frequency are calculated using the relationship of $J'(\omega)$ and $J''(\omega)$ the of Eqs (33) and (34) to $L(\tau)$.

$$J'(\omega) = J_g + \int_{-\infty}^{+\infty} L(\tau) \frac{1}{\omega^2 \tau^2 + 1} \mathrm{dln}\tau \tag{33}$$

$$J''(\omega) = \int_{-\infty}^{+\infty} L(\tau) \frac{\omega\tau}{\omega^2 \tau^2 + 1} \mathrm{dln}\tau \tag{34}$$

The Eqs (33) and (34) are discretized by the chained trapezoidal rule, the integral interval is selected as $[10^{-40}, 10^{30}]$, and the subinterval length is selected as $\log\Delta\rho = 0.005$. The numerical

model expressions of $J'(\omega)$ and $J''(\omega)$ with $L(\tau)$ are respectively:

$$J'(\omega) \approx \frac{1}{10^{\delta+\alpha}} + (\ln 10) \cdot \frac{0.005}{2} \cdot$$

$$\left\{ \begin{array}{l} \dfrac{L\left(10^{-40}\right)}{1 + \omega^2 \left(10^{-40}\right)^2} \\[2em] +2\left[\dfrac{L\left(10^{-39.995}\right)}{1 + \omega^2 \left(10^{-39.995}\right)^2} + \cdots + \dfrac{L\left(10^{29.995}\right)}{1 + \omega^2 \left(10^{29.995}\right)^2}\right] \\[2em] +\dfrac{L\left(10^{30}\right)}{1 + \omega^2 \left(10^{30}\right)^2} \end{array} \right\} \tag{35}$$

$$J''(\omega) \approx \frac{1}{10^{\delta+\alpha}} + (\ln 10) \cdot \frac{0.005}{2} \cdot$$

$$\left\{ \begin{array}{l} L\left(10^{-40}\right)\dfrac{\omega 10^{-40}}{1 + \omega^2 \left(10^{-40}\right)^2} \\[2em] +2\left[L\left(10^{-39.995}\right)\dfrac{\omega 10^{-39.995}}{1 + \omega^2 \left(10^{-39.995}\right)^2} + \cdots + L\left(10^{29.995}\right)\dfrac{\omega 10^{29.995}}{1 + \omega^2 \left(10^{29.995}\right)^2}\right] \\[2em] +L\left(10^{30}\right)\dfrac{\omega 10^{30}}{1 + \omega^2 \left(10^{30}\right)^2} \end{array} \right\} \tag{36}$$

$J'(\omega)$ and $J''(\omega)$ at each reduced frequency are calculated from the continuous retardation spectrum according to Eqs (35) and (36). However, $J'(\omega)$ and $J''(\omega)$ at each reduced frequency calculated from the discrete retardation spectrum can be directly obtained by using Eqs (11) and (12). Then plot the calculated $J'(\omega)$ and $J''(\omega)$ on a graph with master curves, as shown in Fig 19. The master curves of $J'(\omega)$ and $J''(\omega)$ are almost overlapped, and the coefficients of determination $R^2$ are all greater than 0.96. And, $J'(\omega)$ and $J''(\omega)$ obtained from the discrete and continuous retardation spectrum are almost the same as the $R^2$. This shows that the transformation of relaxation spectrum to retardation spectrum in the time domain can obtain a high-precision creep modulus master curves. The creep compliance master curves constructed by discrete and continuous retardation spectrum has very little difference in accuracy.

## 5 Conclusion

In this paper, several viscoelastic parameter master curves were used to comprehensively characterize the mechanical behavior of basalt fiber asphalt mixtures and to analyze the effect of fiber blending on their performance. In particular, master curves for relaxation modulus and delayed flexibility were obtained using discrete and continuous spectral model conversion methods; and, the differences and accuracy of the conversion results between the two methods were compared. The following conclusions were drawn:

1. The addition of basalt fibers improves the strength, deformation resistance and stress relaxation properties of asphalt mixtures, especially the low-temperature properties of the mixture properties. It is worth noting that the improvement of low temperature properties by fibers is mainly achieved by improving the internal structure of asphalt mixtures.

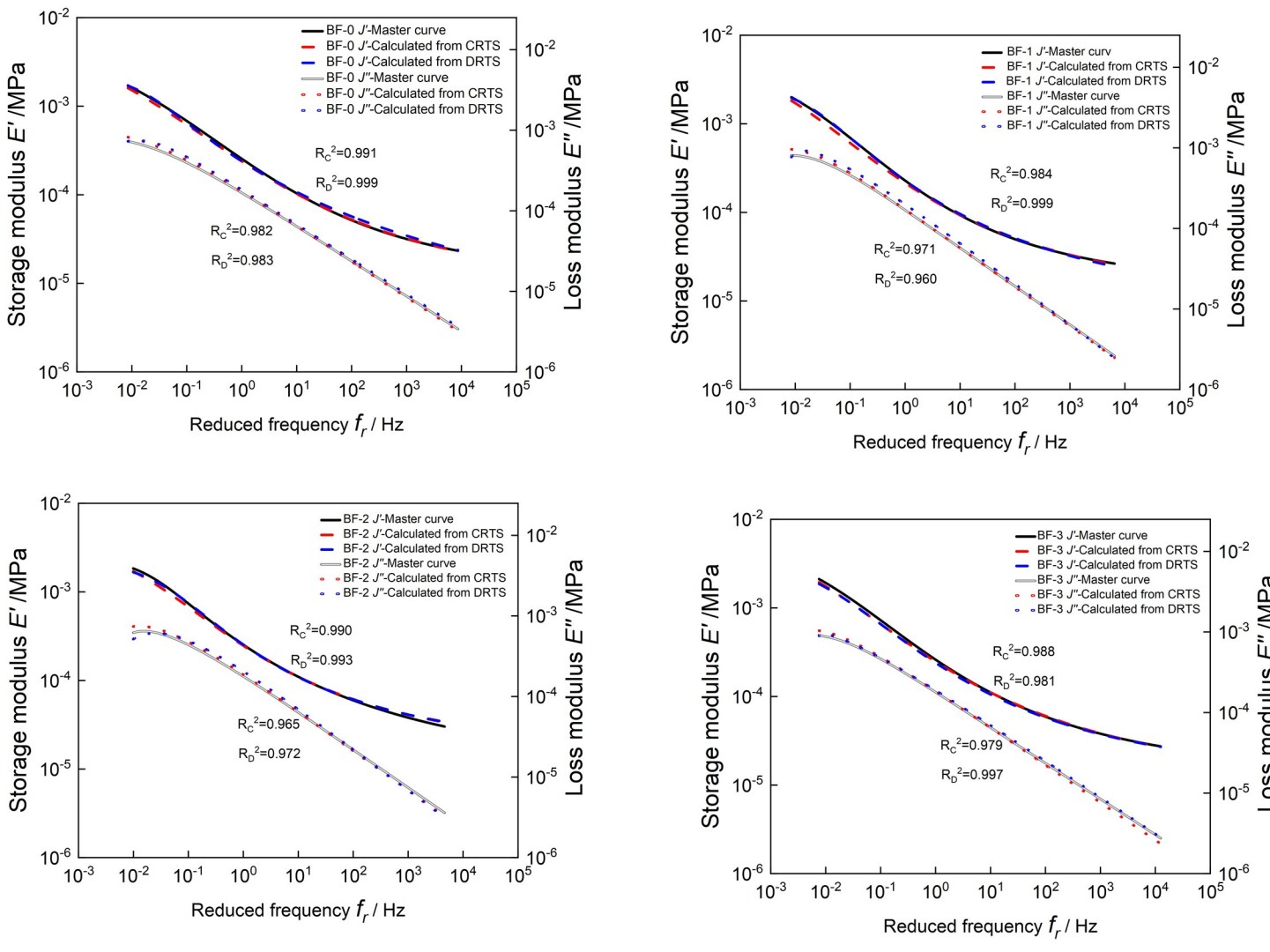

**Fig 19. Comparison of calculated $E'(\omega)$ and $E''(\omega)$ with the master curves of $J'(\omega)$ and $J''(\omega)$.**

2. Asphalt mixtures with a basalt fiber content of 0.2% have the best stress relaxation at low temperatures, as well as better resistance to deformation at high temperatures.

3. Using the K-K relation and the generalized Sigmoidal model, discrete and continuous time spectra containing all the viscoelastic information are obtained from the principal curves of $E'(\omega)$ and $E''(\omega)$ by means of the Prony series model and the Laplace transform.

4. Both the discrete and continuous model conversion methods with 15 discrete time points were selected to obtain highly accurate master curves for relaxation modulus and creep flexibility, and the variability of the conversion results for both was small.

## Author Contributions

**Conceptualization:** Zhuohang Zang.

**Formal analysis:** Xiaogang Kang.

**Investigation:** Qi Huang, Zhengji Zhang.

**Project administration:** Han Yan.

**Resources:** Er-hu Yan.

**Writing – original draft:** Xiaogang Kang.

**Writing – review & editing:** Qi Huang, Pengfei Chen.

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
