## [Decision Letter · Decision Letter 0]

13 Nov 2023

PONE-D-23-33461Characterization of viscoelastic behavior of basalt fiber asphalt mixtures based on discrete and continuous spectrum modelsPLOS ONE

Dear Dr. kang,

Thank you for submitting your manuscript to PLOS ONE. After careful consideration, we feel that it has merit but does not fully meet PLOS ONE’s publication criteria as it currently stands. Therefore, we invite you to submit a revised version of the manuscript that addresses the points raised during the review process.

We look forward to receiving your revised manuscript.

Kind regards,

Jiaolong Ren

Academic Editor

PLOS ONE

Journal Requirements:

Reviewers' comments:

Reviewer's Responses to Questions

**Comments to the Author**

1. Is the manuscript technically sound, and do the data support the conclusions?

Reviewer #1: Yes

Reviewer #2: Yes

Reviewer #3: Yes

2. Has the statistical analysis been performed appropriately and rigorously? 

Reviewer #1: Yes

Reviewer #2: Yes

Reviewer #3: Yes

3. Have the authors made all data underlying the findings in their manuscript fully available?

Reviewer #1: Yes

Reviewer #2: Yes

Reviewer #3: No

4. Is the manuscript presented in an intelligible fashion and written in standard English?

Reviewer #1: Yes

Reviewer #2: Yes

Reviewer #3: Yes

5. Review Comments to the Author

Reviewer #1: This study characterizes the viscoelastic behavior of basalt fiber asphalt mixtures using discrete and continuous spectrum models. Complex modulus tests were conducted on asphalt mixtures with 0%, 0.1%, 0.2%, and 0.3% basalt fiber content. Master curves of dynamic modulus, phase angle, storage modulus, and loss modulus were constructed using the generalized sigmoidal model and approximate Kramers-Kronig relationship. The master curves of relaxation modulus and creep compliance were then determined using discrete Prony series models and continuous spectrum models. Results showed both spectrum models provided highly accurate master curves for relaxation modulus and creep compliance, with minimal differences between them. The addition of basalt fiber was found to enhance strength, stress relaxation, and deformation resistance of the asphalt mixtures.

The topic follows an interesting subject and appreciated work is done. The paper is well written. However, the paper has some minor shortcomings to make it straightforward for the readers, so I will leave my comments as follows:

Comments:

- In the introduction, the paper has skipped explaining a brief background about the subject of the study and relevant literature in the field. Provide a clear background in the first one or two paragraphs.

- “Moreover, many studies characterizing the viscoelasticity…” You have not mentioned any study. Generally, the paper should clearly mention the shortcomings of the past literature to explain its own novelty better.

- More explanation is needed on why the generalized sigmoidal model and approximate Kramers-Kronig relationship were selected.

- I suggest the theoretical background can be shortened. In the material and methods section, you can narrowly focus on the methodology and the research process. So, you can move parts of the formulations to the appendix or just mention them.

- Insert space before reference bracket.

- Overall, the writing is clear, but carefully proofreading to fix minor typos and formatting issues would further polish the manuscript. Some sentences are long and could be broken up or restructured for clarity.

Reviewer #2: 1. need to specify the problem statement clearly

2. Need to identify which layer(surface or binder or base) , the author recommend to use this type of fiber or in general.

3. need to clarify the utilized aspect ration of fiber.

Reviewer #3: The comments are shown in the attached file

It is recommended to check some highlighted phrases in the results and check the number and citation of tables.

In addition, it is highly recommended to cite more recent references.

6. PLOS authors have the option to publish the peer review history of their article (what does this mean?). If published, this will include your full peer review and any attached files.

Reviewer #1: No

Reviewer #2: No

Reviewer #3: No

---

## [Author Response · Author response to Decision Letter 0]

1 Dec 2023

Reply to Reviewer #1

Comment 1:

In the introduction, the paper has skipped explaining a brief background about the subject of the study and relevant literature in the field. Provide a clear background in the first one or two paragraphs.

Response 1:

Thank you for your valuable comments, according to your comments, I have made changes in the introduction, the first paragraph clearly explains the road surface often encountered diseases, and the basalt fiber blending on the asphalt mixture structure improvement, and pointed out that the existing research on the viscoelastic behavior of the problem, more use of dynamic modulus and phase angle analysis of its viscoelastic behavior, rarely a comprehensive study of its viscoelastic properties. The second paragraph points out the conversion problem of the viscoelastic parameters of the mixture, and the third paragraph introduces the role of the time spectrum, and points out the disadvantages of the spectral discretization and the continuous time spectrum to determine the conversion of the main curve of viscoelastic parameters.

Comment 2: 

“Moreover, many studies characterizing the viscoelasticity…” You have not mentioned any study. Generally, the paper should clearly mention the shortcomings of the past literature to explain its own novelty better.

Response 2:

In response to your comments, we have added some references in the first paragraph of the introduction that point out problems in existing research.

Comment 3:

More explanation is needed on why the generalized sigmoidal model and approximate Kramers-Kronig relationship were selected.

Response 3: 

Thank you for your comment, I explained in lines 65-71 that the approximate Kramers-Kronig (K-K) relationship can model the dynamic modulus and phase angle master curves, the storage modulus and loss modulus master curves, and can represent the interrelationships of the viscoelastic parameters well, and that it is a simple and convenient way to represent the loss modulus master curves by using the approximate K-K relationship between the storage modulus and loss modulus process. The asphalt mixture master curves established by the Sigmoidal Model (GSM), which is capable of quasi-taking the dynamic mechanical response with high correlation coefficients and accurately predicting the dynamic mechanical properties, are clearly explained in lines 71-82. Simultaneous application of the approximate K-K relation and the GSM allows the construction of master curves that both ensure that the plotted master curve conforms to the line viscoelasticity and allow the master curve not to be symmetric about the viewpoint.

Comment 4: 

I suggest the theoretical background can be shortened. In the material and methods section, you can narrowly focus on the methodology and the research process. So, you can move parts of the formulations to the appendix or just mention them.

Response 4: 

Thank you for your comments, the background of this article accordingly made deletions, the article only retained the important formulas, the derivation process of the formula by the formula to text, so that the reader can make a clearer understanding of the process. In addition to this, in order to make the paper more readable and professional and to bring out the theme of viscoelasticity and fibers. Therefore, we have cut down the formulas that unnecessarily show the derivation process and made necessary adjustments to the syntax of the article. These adjustments do not affect the logic and science of the original paper.

Comment 5: 

Insert space before reference bracket.

Response 5: 

Thank you for your detailed review. We have carefully revised and reviewed this article for such problems as you mentioned.

Comment 6: 

Overall, the writing is clear, but carefully proofreading to fix minor typos and formatting issues would further polish the manuscript. Some sentences are long and could be broken up or restructured for clarity.

Response 6: 

Thank you for your comments, we have carefully and thoroughly proofread the manuscript, corrected all grammar and typos, and split longer sentences.

Reply to Reviewer #2

Comment 1:

 need to specify the problem statement clearly.

Response 1:

Thank you for your valuable comments, according to your comments I have revised and adjusted the background in the introduction, now the introduction can be more clear description of the research background of this paper as well as the focus of the research.

Comment 2: 

Need to identify which layer(surface or binder or base) , the author recommend to use this type of fiber or in general.

Response 2: 

Line 95 describes that basalt fiber asphalt mixtures are commonly used in the upper and middle layers of asphalt pavements; lines 98-99 describe the type of material and source of basalt fibers, "Basalt fibers are selected from 9mm short-cut basalt fibers produced by Changsha North American Float Company."

Comment 3:

need to clarify the utilized aspect ration of fiber.

Response 3:

Line 100 of the article states that basalt fibers have an L/D ratio of about 560.

Reply to Reviewer #3

Comment 1: 

The comments are shown in the attached file.

Response 1:

Thank you for your detailed comments, this paper has been carefully and meticulously revised according to the labeling information you attached to the errors in the article, such as the references of tables and figures in the text, as well as the description of the graphs and charts. Among them, for Fig. 2 and Fig. 3 marked in your article, it is further explained to you that the English terminology to express the frequency reduction is not a problem, and these two diagrams represent the phase angle and dynamic modulus of basalt fibers with different dopant amounts at different shrinkage frequencies, respectively, and then plotted as a master curve, so there is no error in the title and the transverse coordinates.

Comment 2: 

It is recommended to check some highlighted phrases in the results and check the number and citation of tables.

Response 2:

Thank you for your detailed comments, I have changed the incorrect information in the conclusion labeling and have double checked and corrected the citations of the charts and added citations where they were missing and they are now all cited correctly and clearly. 

Comment 3: 

In addition, it is highly recommended to cite more recent references.

Response 3:

Based on your comments, I have included references to the latest articles.

We would like to take this opportunity to thank you for all your time involved and this great opportunity for us to improve the manuscript. We hope you will find this revised version satisfactory.

---

## [Decision Letter · Decision Letter 1]

6 Dec 2023

Characterization of viscoelastic behavior of basalt fiber asphalt mixtures based on discrete and continuous spectrum models

PONE-D-23-33461R1

Dear Dr. Yan,

We’re pleased to inform you that your manuscript has been judged scientifically suitable for publication and will be formally accepted for publication once it meets all outstanding technical requirements.

Kind regards,

Jiaolong Ren

Academic Editor

PLOS ONE

Additional Editor Comments (optional):

Reviewers' comments:

Reviewer's Responses to Questions

**Comments to the Author**

1. If the authors have adequately addressed your comments raised in a previous round of review and you feel that this manuscript is now acceptable for publication, you may indicate that here to bypass the “Comments to the Author” section, enter your conflict of interest statement in the “Confidential to Editor” section, and submit your "Accept" recommendation.

Reviewer #1: All comments have been addressed

Reviewer #2: All comments have been addressed

Reviewer #3: (No Response)

2. Is the manuscript technically sound, and do the data support the conclusions?

Reviewer #1: Yes

Reviewer #2: Yes

Reviewer #3: Yes

3. Has the statistical analysis been performed appropriately and rigorously? 

Reviewer #1: Yes

Reviewer #2: Yes

Reviewer #3: Yes

4. Have the authors made all data underlying the findings in their manuscript fully available?

Reviewer #1: Yes

Reviewer #2: Yes

Reviewer #3: Yes

5. Is the manuscript presented in an intelligible fashion and written in standard English?

Reviewer #1: Yes

Reviewer #2: Yes

Reviewer #3: Yes

6. Review Comments to the Author

Reviewer #1: Dear Authors,

I would like to express my appreciation for your diligent work in addressing the concerns raised during the review process. Your efforts have significantly improved the quality of the manuscript, and it is now a strong contribution to our field.

Best regards,

Reviewer #2: Good work and the most important notes have been done, need to complete the requirement of the journal

Reviewer #3: (No Response)

7. PLOS authors have the option to publish the peer review history of their article (what does this mean?). If published, this will include your full peer review and any attached files.

Reviewer #1: No

Reviewer #2: No

Reviewer #3: **Yes: **Abeer Khudhur Jameel

---

## [Editor Report · Acceptance letter]

11 Jan 2024

PONE-D-23-33461R1 

PLOS ONE

Dear Dr. Yan, 

I'm pleased to inform you that your manuscript has been deemed suitable for publication in PLOS ONE. Congratulations! Your manuscript is now being handed over to our production team.

Kind regards, 

on behalf of

Dr. Jiaolong Ren 

Academic Editor

PLOS ONE